# Recognition of phylogenetically diverse pathogens through enzymatically amplified recruitment of RNF213

Ana Crespillo-Casado[1,5], Prathyush Pothukuchi[1,5], Katerina Naydenova[1], Matthew C J Yip [1], Janet M Young [2], Jerome Boulanger[1], Vimisha Dharamdasani[1], Ceara Harper[1], Pierre-Mehdi Hammoudi[1], Elsje G Otten[1], Keith Boyle [1], Mayuri Gogoi [1], Harmit S Malik[2,3] & Felix Randow [1,4]✉

## Abstract

**Innate immunity senses microbial ligands known as pathogen-associated molecular patterns (PAMPs). Except for nucleic acids, PAMPs are exceedingly taxa-specific, thus enabling pattern recognition receptors to detect cognate pathogens while ignoring others. How the E3 ubiquitin ligase RNF213 can respond to phylogenetically distant pathogens, including Gram-negative *Salmonella*, Gram-positive *Listeria*, and eukaryotic *Toxoplasma*, remains unknown. Here we report that the evolutionary history of RNF213 is indicative of repeated adaptation to diverse pathogen target structures, especially in and around its newly identified CBM20 carbohydrate-binding domain, which we have resolved by cryo-EM. We find that RNF213 forms coats on phylogenetically distant pathogens. ATP hydrolysis by RNF213's dynein-like domain is essential for coat formation on all three pathogens studied as is RZ finger-mediated E3 ligase activity for bacteria. Coat formation is not diffusion-limited but instead relies on rate-limiting initiation events and subsequent cooperative incorporation of further RNF213 molecules. We conclude that RNF213 responds to evolutionarily distant pathogens through enzymatically amplified cooperative recruitment.**

**Keywords** Host-pathogen Interaction; Pattern Recognition Receptor; Innate Immunity; Positive Selection; Ubiquitylation
**Subject Categories** Microbiology, Virology & Host Pathogen Interaction; Post-translational Modifications & Proteolysis; Structural Biology

## Introduction

The innate immune system deploys pattern recognition receptors (PRRs) to detect pathogens. PRRs are germline-encoded sensors for microbial ligands known as pathogen-associated molecular patterns (PAMPs). PAMPs are evolutionarily conserved molecules produced by most species in a given taxon, such as lipopolysaccharide in Gram-negative bacteria (Janeway, 1989). Therefore, while PRRs can detect exposure to most species from large taxonomic groups, they are necessarily limited to sensing cognate pathogens while ignoring others. Receptors for nucleic acids are a noticeable exception due to the much wider distribution of their ligands (Tan et al, 2015); however, nucleic acids are usually shielded from detection by PRRs inside pathogens. Recently, the E3 ligase RNF213 has emerged as a restriction factor for phylogenetically diverse pathogens. We initially discovered that RNF213 ubiquitylates LPS on *Salmonella enterica* serovar Typhimurium (*S.* Typhimurium), which represented the first known instance of ubiquitylation targeting a non-proteinaceous biomolecule, and labels bacteria as targets for autophagy (Otten et al, 2021). Since this discovery, it has become evident that RNF213 is not specific for Gram-negative bacteria; it also senses the Gram-positive bacterium *Listeria monocytogenes* (*L. monocytogenes*), the apicomplexan parasite *Toxoplasma gondii* (*T. gondii*), as well as viruses (Thery et al, 2021; Hernandez et al, 2022; Houzelstein et al, 2021; Walsh et al, 2022; Matta et al, 2023). However, how RNF213 detects such phylogenetically diverse pathogens remains unclear.

RNF213, also known as mysterin, is the largest ubiquitin E3 ligase in the human genome. It was originally identified as a susceptibility gene for Moyamoya disease, a progressive stenosis of the inner carotid artery, which leads to compensatory growth of abnormal collateral vessels that appear in angiograms as a 'puff of smoke' (= Moyamoya in Japanese) (Kamada et al, 2011; Liu et al, 2011). The incidence of Moyamoya disease is highest in South-East Asia due to a founder-effect missense mutation (RNF213$_{R4810K}$). However, the penetrance of the RNF213$_{R4810K}$ allele is less than 1%,

[1]MRC Laboratory of Molecular Biology, Division of Protein and Nucleic Acid Chemistry, Francis Crick Avenue, Cambridge CB2 0QH, UK. [2]Division of Basic Sciences, Fred Hutchinson Cancer Research Center, Seattle, WA, USA. [3]Howard Hughes Medical Institute, Fred Hutchinson Cancer Research Center, Seattle, WA, USA. [4]University of Cambridge, Department of Medicine, Addenbrooke's Hospital, Cambridge CB2 2QQ, UK. [5]These authors contributed equally: Ana Crespillo-Casado, Prathyush Pothukuchi. ✉E-mail: randow@mrc-lmb.cam.ac.uk

suggesting that additional genetic or environmental factors, for example infections, may contribute to Moyamoya disease (Ihara et al, 2022; Asselman et al, 2022).

Unusual for E3 ubiquitin ligases, RNF213 possesses two unrelated ligase domains, an RZ finger and a RING domain (Otten et al, 2021). Although the isolated RING domain has been shown to function in vitro, the RZ finger catalyzes LPS- and auto-ubiquitylation without apparent contribution from the RING domain (Otten et al, 2021; Ahel et al, 2021; Takeda et al, 2020; Bhardwaj et al, 2023; Habu and Harada, 2021). Unique amongst E3 ligases, *RNF213* also encodes a dynein-like arrangement of six AAA+ ATPase domains, two of which contain Walker A and B motifs and are catalytically active (Ahel et al, 2020). ATP hydrolysis has been suggested to foster access of RNF213 to inner LPS structures otherwise hidden by the O-antigen layer, but the precise function of the ATPase ring remains unknown (Otten et al, 2021).

Intracellular pathogens inhabit specific cellular compartments, reflecting the pathogen's need for host factors and nutrients as well as its ability to neutralize compartment-specific host defence mechanisms (Petit and Lebreton, 2022; Ray et al, 2009). For example, *Salmonella* Typhimurium is a facultatively intracellular Gram-negative bacterium adapted to life inside bacteria-containing vacuoles. However, a considerable proportion of *S.* Typhimurium damages the limiting membrane of their vacuoles and gains access to the host cytosol (Knodler, 2015). In contrast, the Gram-positive bacterium *L. monocytogenes* actively and efficiently escapes from phagosomes into the cytosol, a location to which it is well adapted as evidenced by its actin-dependent intra- and intercellular motility. *L. monocytogenes ΔactA*, lacking the actin-assembly protein ActA, becomes immotile and prone to autophagy (Yoshikawa et al, 2009; Kocks et al, 1995; Domann et al, 1992). Finally, *Toxoplasma gondii*, an apicomplexan parasite, is an obligate intracellular pathogen that actively invades host cells in a rapid and complex process, resulting in the formation of a parasitophorous vacuole (Bisio and Soldati-Favre, 2019). The parasitophorous vacuole membrane keeps the parasite separated from the cytosol, thus avoiding any direct attack by cytosolic defence mechanisms. However, the parasitophorous vacuole membrane itself can become a target for cell-autonomous immunity. For example, both murine and human cells ubiquitylate substrates in the *T. gondii* parasitophorous vacuole membrane, especially in interferon (IFN)-stimulated cells, despite differences in the repertoire of restriction factors, such as the prominent role of IRGs in mice (Clough and Frickel, 2017; Matta et al, 2021; Mukhopadhyay et al, 2020).

Here, we investigated how RNF213 attacks phylogenetically distant pathogens that inhabit different sub-cellular compartments: *S.* Typhimurium, *L. monocytogenes*, and *T. gondii*. To aid our analyses, we solved the structure of human RNF213 and discovered a novel carbohydrate-binding CBM20 domain in the previously unresolved N-terminus. We found that the CBM20 domain and adjacent parts of the stalk region are under intense positive selection in simian RNF213 genes, indicating that direct interactions between RNF213 and variable pathogen ligands may have shaped RNF213 function in an evolutionary arms race. We found that RNF213 forms dense protein coats on *S.* Typhimurium and *L. monocytogenes* after they have invaded the host cytosol. In contrast, in cells infected with *T. gondii* RNF213 coats form on the cytosolic face of the parasitophorous vacuole membrane. Unexpectedly, we

find that coats are not created through diffusion-limited recruitment of RNF213 monomers but rather require rare, rate-limiting initiation events, from which coats grow in multiple directions through cooperative incorporation of new RNF213 subunits. Coat formation on all three pathogens requires AAA+ ATPase activity, whereas RZ finger-mediated E3 ligase activity is only needed for coat formation on bacteria. We conclude that RNF213 responds to evolutionarily distant pathogens through an enzymatically amplified cooperative recruitment mechanism that establishes dense RNF213 coats on target structures, including bacterial surfaces and the membrane of host-derived vacuoles encasing eukaryotic parasites.

## Results

### RNF213 senses phylogenetically distant pathogens

To investigate the ability of RNF213 to detect phylogenetically distant pathogens, we infected cells with the Gram-negative bacterium *S.* Typhimurium, the Gram-positive bacterium *L. monocytogenes*, or the apicomplexan parasite *T. gondii*. For the latter, we used both the type I strain RH and the type II strain Pru, which differ in their virulence and ability to antagonize host defences (Mukhopadhyay et al, 2020). In both human epithelial cells and in RNF213[KO] MEFs complemented with GFP-tagged human RNF213, we found that RNF213 was recruited to all three pathogens (Figs. 1A,B and EV1A). In cells infected with *S.* Typhimurium or *L. monocytogenes ΔactA*, RNF213 accumulated on individual bacteria of well-established micro-colonies (Fig. 1A), indicating RNF213 recruitment to the surface of bacteria after they had escaped from the bacteria-containing vacuole; these findings are consistent with previous reports (Otten et al, 2021; Thery et al, 2021). In contrast, in cells infected with *T. gondii*, RNF213 was not recruited to the surface of the parasite itself but rather accumulated on the membrane of the parasitophorous vacuole, as revealed by the continuous RNF213 coat encircling two parasites (Fig. 1A).

Since recruitment of GFP-RNF213 to all pathogens under investigation coincided with a dense ubiquitin coat (Fig. 1A), we tested whether endogenous RNF213 is required for ubiquitin deposition. Indeed, we found that ubiquitylation of *L. monocytogenes ΔactA* relied entirely on RNF213 (Fig. 1C,D), similar to the situation in cells infected with *S.* Typhimurium (Otten et al, 2021). In cells infected with the Pru strain of *T. gondii*, parasitophorous vacuoles were readily ubiquitylated, particularly in cells pre-treated with IFNγ, in contrast to cells infected with the RH strain, where vacuoles remained largely ubiquitin negative (Figs. 1E and EV1B). Based on these observations and previous reports (Otten et al, 2021; Thery et al, 2021; Matta et al, 2023; Hernandez et al, 2022), we conclude that RNF213 targets phylogenetically distant pathogens that can either be cytosol-exposed, such as *S.* Typhimurium and *L. monocytogenes*, or shielded by a host-derived membrane, such as *T. gondii*.

### Formation of the RNF213 coat is a two-stage process

To investigate how the RNF213 coat develops on phylogenetically distant pathogens we performed time-resolved live cell microscopy. We observed that on vacuoles containing *T. gondii*, RNF213 coats

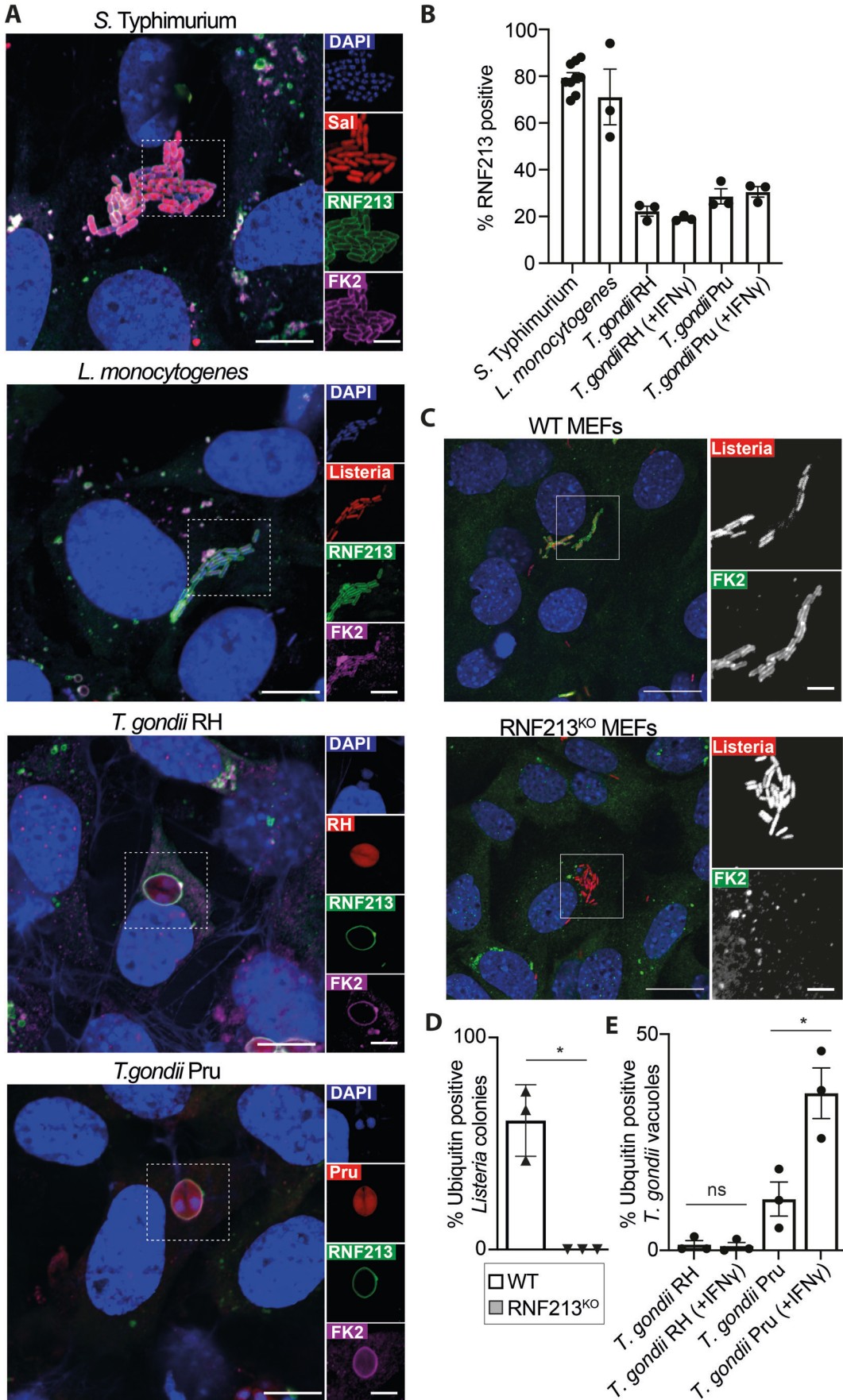

◄ **Figure 1. RNF213 recruitment to phylogenetically distant pathogens.**

(A) Confocal micrographs of RNF213$^{KO}$ MEFs stably expressing GFP-RNF213, stained with anti-ubiquitin (FK2) antibody and DAPI. Cells were fixed at 4 h post-infection with mCherry-expressing *S. Typhimurium*, 6 h post-infection with mCherry-expressing *L. monocytogenes ΔActA* and 24 h post-infection with Tomato-expressing *T. gondii* Type I strain RH or Type II strain Pru. Regions marked with white borders in the main images are shown magnified on the right. Scale bar, 10 μm (magnification box scale bar, 5 μm). (B) Percentage of cytosolic *S. Typhimurium*, *L. monocytogenes ΔActA* colonies or *T. gondii* vacuoles positive for GFP-RNF213 at 4 h, 6 h or 1 h p.i., respectively, in RNF213$^{KO}$ MEFs stably expressing GFP-RNF213 treated or not with IFNγ. Mean $+/-$ SEM of $n = 9$ (*S. Typhimurium*), $n = 3$ (*L. monocytogenes*) and $n = 3$ (*T. gondii*) independent biological experiments, each performed as technical triplicates and assessing $n > 100$ bacteria and $n > 250$ parasites per experiment. (C) Representative confocal micrographs of WT and RNF213$^{KO}$ MEFs infected with mCherry-expressing *L. monocytogenes ΔActA* and stained at 6 h p.i. with anti-ubiquitin (FK2) antibody and DAPI. Columns on the right represent magnified regions marked with white borders in main image. Scale bar, 20 μm (magnification box scale bar, 5 μm). (D) Percentage of *L. monocytogenes ΔActA* colonies positive for ubiquitin (FK2) at 6 h post-infection in WT and RNF213$^{KO}$ MEFs. Mean $+/-$ SEM of $n = 3$ independent biological experiments, each performed as technical triplicates. $n > 100$ colonies per experiment. Two-tailed paired Student t-test, $*p = 0.0247$. (E) Percentage of *T. gondii* Type I RH strain or Type II Pru strain positive for ubiquitin (FK2) at 1 h post-infection in WT MEFs treated or not with IFNγ. Mean $+/-$ SEM of $n = 3$ independent biological experiments, each performed in technical triplicates and assessing $n > 250$ parasites per experiment. Two-tailed paired Student t-test, $*p = 0.0098$. Source data are available online for this figure.

were initiated at a single point, before growing in multiple directions until the majority of the parasitophorous vacuole was covered (Movie EV1, Fig. 2A,B). To further study the kinetics of coat formation, we monitored the intensity of the GFP-RNF213 signal over time around the parasitophorous vacuole using a Fiji script. From the fitted curve we defined the beginning, mid- and endpoint of coat formation as 5%, 50% and 95% of maximum RNF213 recruitment (illustrated in Fig. 2C for the parasite shown in Fig. 2A). The analysis of 27 parasites revealed that coat formation started at variable time points throughout the observation period (which lasted from 20 min to 120 min post-infection) (Figs. 2D and EV2). However, once coat formation had been initiated, coats expanded swiftly and were completed within $5.0 \pm 2.4$ min (median ± MAD) (Fig. 2E). To further analyse coat initiation, we plotted the fraction of observed parasites that remained uncoated as a function of time (Fig. 2F). Modelling coat initiation as a Poisson point process, we derived a time constant $T = 27 \pm 1$ min, which represents the average 'waiting' time for coat initiation. The difference in the time constant of coat initiation and the time required for coat expansion explains why coats on *T. gondii* vacuoles are initiated from a single focus; rarity of coat initiation and rapid coat completion limit the likelihood of more than one initiation event occurring per vacuole. Based on these observations, we conclude that coat formation on *T. gondii* is not a simple diffusion-limited process of ligand-receptor interactions. Rather, coat formation occurs in two distinct steps: following a rare and rate-limiting initiation event, possibly involving a single RNF213 molecule only, coats subsequently grow rapidly through cooperative incorporation of further RNF213 molecules (Fig. 2B).

Similarly, we find that coat formation on *L. monocytogenes ΔactA* is also not diffusion-limited, as rate-limiting initiation events and rapid cooperative expansion phases were observed (Movie EV2, Fig. 2G,H). However, in contrast to *T. gondii* and *S. Typhimurium* (Otten et al, 2021), coats on *L. monocytogenes ΔactA* originate in multiple locations around a single bacterium. This difference in single- versus multi-focus initiation of RNF213 coats is likely caused by differences in the identity and concentration of coat-initiating ligand(s) or corresponding ligand-binding domains in RNF213. Taken together, we conclude that, for all pathogens under study, RNF213 coats are formed through rate-limiting initiation events and subsequent cooperative growth. We propose that both coat initiation and coat expansion require receptor-mediated pathogen sensing and that an additional force,

for example RNF213 oligomerization, causes the preferential incorporation of new monomers in the vicinity of already bound RNF213 molecules.

## Strong positive selection has shaped the RNF213 N-terminus

Since RNF213 accumulates on the surface of both Gram-negative and Gram-positive bacteria as well as on the parasitophorous vacuole surrounding the eukaryotic parasite *T. gondii*, the question arises of how RNF213 detects such phylogenetically diverse pathogens. If RNF213 recruitment were mediated by direct binding to PAMPs, an evolutionary arms race for binding affinity between RNF213 and diverse pathogens may have ensued, resulting in signatures of ligand-imposed positive selection in RNF213. In contrast, if RNF213 recruitment was indirect, i.e. mediated by hypothetical upstream pattern-recognition receptors, we would not expect to find signatures of recurrent positive selection on RNF213 (Daugherty and Malik, 2012). We therefore analysed simian primate orthologs of RNF213 using a maximum likelihood algorithm that compares the rates of non-synonymous and synonymous substitutions (dN/dS) to search for signatures of positive selection (see Methods). We observed statistically significant positive selection ($p < 10^{-40}$) with high confidence signals (BEB $\geq 0.9$) at 59 of the 5241 aligned codons (Fig. 3A, Dataset EV2). This finding of very strong positive selection in RNF213 is consistent with direct interactions of RNF213 with variable pathogen-derived ligands. These rapidly evolving sites are scattered through multiple domains of RNF213, including the E3 module and the dynein-like AAA+ ring. However, positively selected sites are particularly concentrated in the stalk domain and N-terminal regions that had remained unresolved in the previously published structure of murine RNF213 (Ahel et al, 2020), with 27 of the N-terminal 1000 codons having undergone positive selection.

## A CBM20 carbohydrate-binding domain in the N-terminus of RNF213

To gain further insight into RNF213 functionality, we solved the structure of the human protein to 3 Å resolution by single particle cryo-EM (Figs. 3B–D and EV3C–F; Appendix Table S2). Similar to murine RNF213 (Ahel et al, 2020), we find that the human orthologue appeared as a monomer with an ATP bound to its catalytically inactive

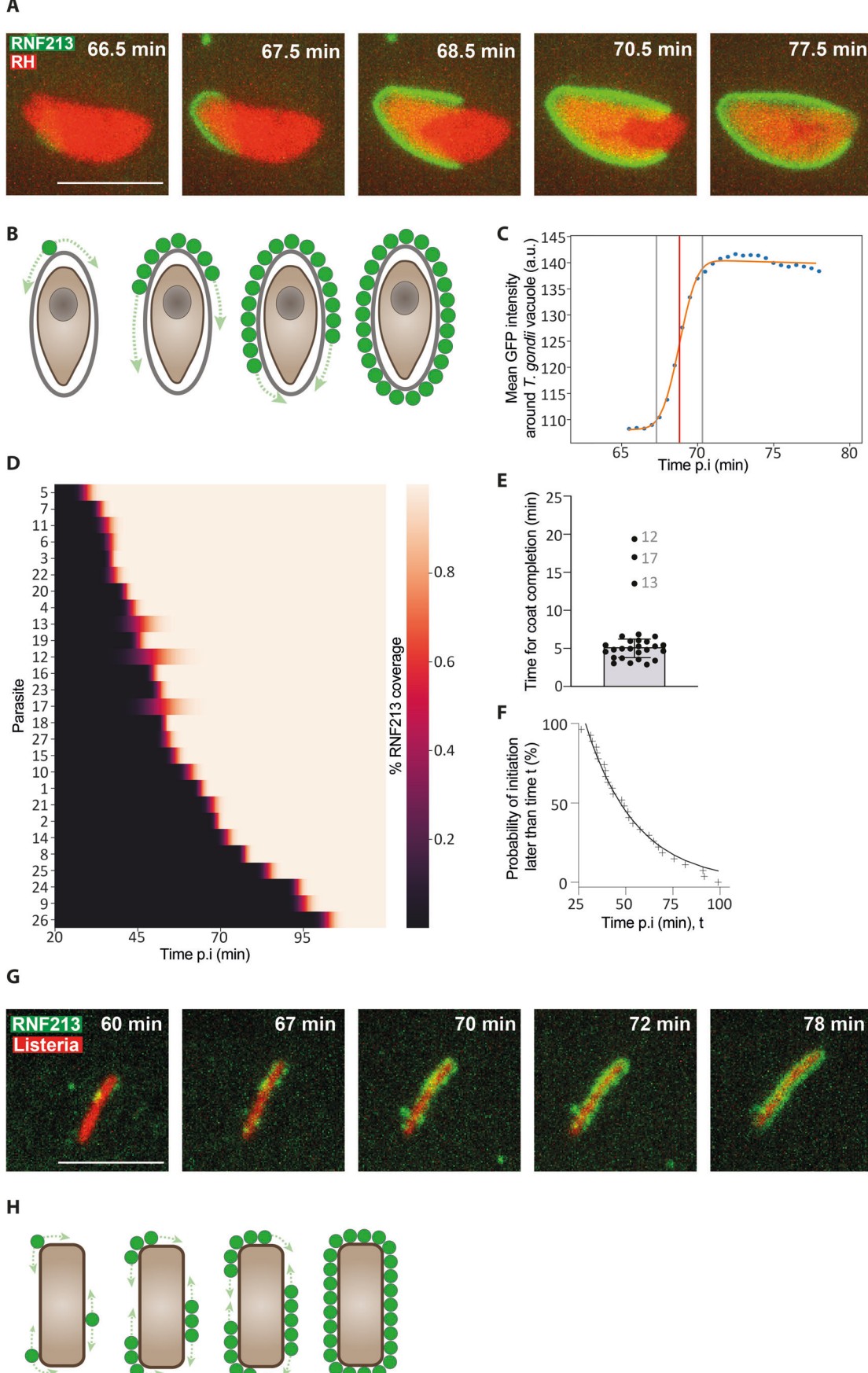

**Figure 2. RNF213 coats form through rate-limiting initiation events and cooperative expansion.**

(A) Still images from Movie EV1. Instant structured illumination microscopy (I-SIM) of RNF213[KO] MEFs stably expressing GFP-RNF213 and infected with Tomato-expressing *T. gondii* Type I RH strain. Times after infection are as indicated. Scale bar, 5 μm. (B) Model for the formation of RNF213 coats on the parasitophorous vacuole of *T. gondii*. (C) Kinetics of GFP-RNF213 accumulation on a *T. gondii* vacuole. Data extracted from parasite 2 in Movie EV1 (also shown in Fig. 2A). Dots represent the measured GFP intensity. An error function was fitted (orange line) to calculate the 5% and 95% of maximum coat formation (vertical grey lines) as well as the midpoint (vertical red line). (D) Heatmap of RNF213 coat formation on 27 *T. gondii* parasites over time. Percentage of RNF213 coverage (low to high) is represented as a colour gradient (black to white). Parasites ordered by midpoint of coat formation. *n* = 6 biological experiments; parasites 1 and 2 from experiment 1; parasites 3 and 4 from experiment 2; parasites 5 to 10 from experiment 3; parasite 11 from experiment 4; parasites 12 to 21 from experiment 5; parasites 22 to 27 from experiment 6. Primary data for all parasites in Fig. EV2. Parasites 1 and 2 are also shown in Movie EV1, parasite 2 in Fig. 2A,C. (E) Time required for coat formation of the 27 parasites shown in Fig. 2D. Outliers are numbered; Median ± MAD (mean absolute deviation). (F) Likelihood of RNF213 coat formation on *T. gondii* parasitophorous vacuoles. '+' symbols represent the initiation time point of coat formation, equivalent to 5% of maximal GFP-RNF213 intensity from the per-parasite coating curve fits (Fig. EV2). The black line represents the probability distribution P(T_initiation > t) = exp(−(t−t0)/T) for the time of first arrival of an event obeying Poisson statistics, where t0 is an arbitrary time offset of the start of the observation and T the time constant of the Poisson process (see Methods). The fit describes the observation with $R^2$ = 98% for T = 27 ± 1 min. (G) Still images from Movie EV2. Instant structured illumination microscopy (I-SIM) of RNF213[KO] MEFs stably expressing GFP-RNF213 and infected with mCherry-expressing *L. monocytogenes ΔActA*. Scale bar, 5 μm. (H) Model for the formation of RNF213 coats on *L. monocytogenes*. Source data are available online for this figure.

AAA2 ATPase domain. Accounting for the flexibility between the individual domains of RNF213, the murine and human structures can be flexibly superimposed with mean Cα deviation of less than 2.1 Å, indicating an overall conserved protein fold. Focused local refinement of the previously unresolved N-terminus revealed the existence of a novel domain (amino acids 378 to 515) with structural similarity to the CBM20 carbohydrate-binding module of glucoamylase from *Aspergillus niger* (PDB 1AC0) (Sorimachi et al, 1997). CBM20 domains occur in a variety of microbial and eukaryotic carbohydrate-active enzymes (CAZymes) and are characterized by a beta-sandwich core that supports two topologically separate carbohydrate-binding sites, known as site 1 and site 2 (Garron and Henrissat, 2019). Due to the flexibility of the CBM20 domain relative to the stalk, the CBM20 domain in human RNF213 is resolved at a local resolution of ~4.5 Å. We therefore created an AlphaFold (Jumper et al, 2021) model of the region of interest (Fig. 3E,F), which, in agreement with the cryo-EM density, predicts the existence of a beta-sandwich structure in the domain and thus supports our experimental evidence for the existence of a bona fide CBM20 domain in RNF213.

We next mapped the position of positively selected residues onto the structure of human RNF213 and found that they are mainly positioned at the surface of the protein (Fig. 3G). A large cluster of 23 positively selected residues covers the CBM20 domain and the exterior face of the adjacent stalk region. We conclude that the surface of RNF213, particularly the CBM20 domain and the adjacent parts of the stalk region, has undergone strong positive selection, a finding consistent with microbial ligands having driven the evolution of RNF213 and suggestive of RNF213 serving as a bona fide pattern recognition receptor.

To investigate the potential contribution of the positively selected N-terminus to pathogen detection, we generated N-terminally truncated deletion mutants (Fig. 3H–J). RNF213_{ΔN367}, lacking the extreme N-terminus but maintaining the CBM20 domain, was recruited efficiently to *S.* Typhimurium and *L. monocytogenes* but was partially impaired in targeting the *T. gondii* parasitophorous vacuole, suggesting that the N-terminal 367 amino acids of RNF213 contribute to the efficient recognition of *T. gondii* vacuoles. RNF213_{ΔN586}, lacking the CBM20 domain and adjacent parts of the stalk, was no longer recruited to any pathogen. However, its lack of recruitment coincided with the formation of bright structures in cells expressing GFP-RNF213_{ΔN586}, which may sequester RNF213 and thereby impair coat formation non-specifically (Fig. 3K). We therefore tested whether point mutations in residues of the CBM20 domain

predicted to mediated carbohydrate binding would interfere with RNF213 coat formation (Appendix Fig. S1). None of the alleles tested affected recruitment of RNF213 to *S.* Typhimurium, *L. monocytogenes* or *T. gondii*, an ambiguous result that fails to prove a role for the CBM20 domain in pathogen recognition but does not rule out its contribution either.

## Enzymatic activity in RNF213 required for coat formation on pathogens

To elucidate the requirement of enzymatic activity in RNF213 for coat formation, we created RNF213 alleles with altered AAA+ ATPase or E3 ligase domains. RNF213 contains a central dynein-like module comprised of two catalytically active and four inactive AAA+ domains (Fig. 3B). We mutated the Walker A and B motifs in the two catalytically active AAA+ domains to disable ATP binding (RNF213_{K2426A}, RNF213_{K2775A}) or ATP hydrolysis (RNF213_{E2488A}, RNF213_{E2845A}), respectively (Fig. 4A,B). None of the four RNF213 mutants coated *L. monocytogenes ΔactA* or *T. gondii* vacuoles, similar to the situation in *S.* Typhimurium (Otten et al, 2021). We conclude that ATP-binding and -hydrolysis by the dynein-like AAA+ module is a specific requirement to form RNF213 coats on evolutionarily distant pathogens, possibly through effects on RNF213 oligomerization.

Unique amongst known E3 ubiquitin ligases, RNF213 possesses two ligase domains: a conventional RING finger and the recently identified RZ domain (Fig. 3B) (Otten et al, 2021; Ahel et al, 2020). To investigate the role of the ligase domains in pathogen recognition and ubiquitylation, we generated RNF213 alleles with defects in either domain. To incapacitate the RING domain, we either removed the entire domain or introduced a point mutation (H4014N), found in a patient with Moyamoya disease (Guey et al, 2017) and predicted to disrupt $Zn^{2+}$ coordination. To incapacitate the RZ domain, we took advantage of our cryo-EM structure of human RNF213 (Fig. 4C). The RZ finger is connected to the E3 module via linker peptides that provide the domain with significant flexibility, thus explaining its apparent absence from the previously published murine RNF213 model and its relatively poor resolution in the human RNF213 structure. To further aid our analyses of the RZ finger, we generated an AlphaFold (Jumper et al, 2021) model (Fig. 4C). Aided by the model, the cryo-EM structure of RNF213 reveals that the RZ domain is comprised of a short alpha helix and two beta-sheets, each comprising two anti-parallel beta-strands. The resulting beta sandwich structure is stabilized through a bound

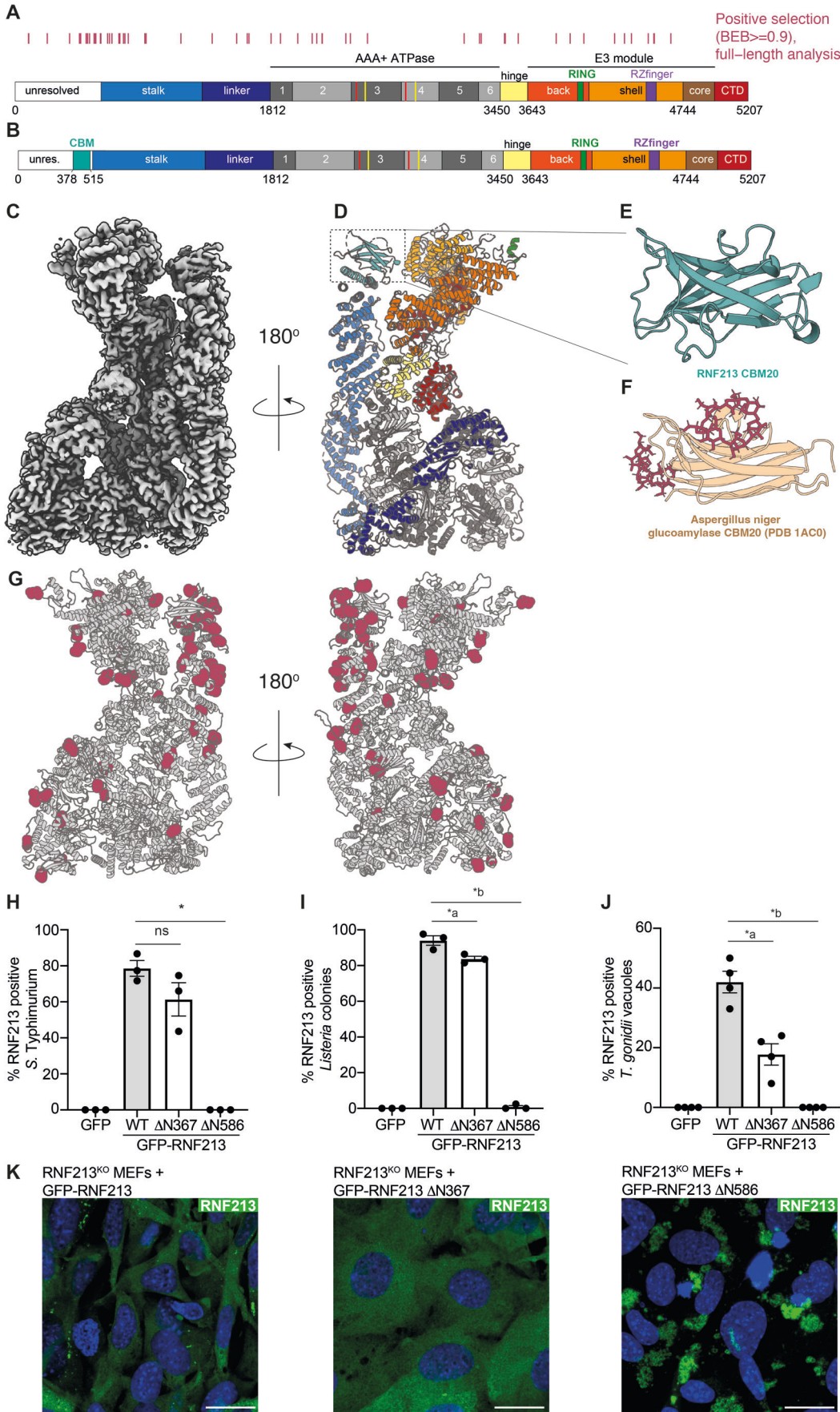

© The Author(s)

◀ **Figure 3. The CBM20 domain and the adjacent stalk region in the N-terminus of RNF213 are under strong positive selection.**

(A) Residues with a high probability (Bayes Empirical Bayes posterior probability ≥ 0.9) of having evolved under positive selection among 24 simian primate species are indicated by red tick marks above a cartoon of RNF213. Numbers refer to human amino acids mapped onto domain boundaries obtained from the murine RNF213 structure (PDB 6TAX). Critical residues in the Walker A (K2426 and K2775) and Walker B motifs (E2488 and E2845) of catalytically active AAA+ ATPase domains are highlighted with red and yellow lines, respectively. Results are similar even if the alignment is split into segments that are free from any evidence of recombination (Fig. EV3A,B, Appendix Table S1). (B) Cartoon representation of human RNF213 domains, using the human structure (PDB 8S24) as reference. The newly solved CBM20 domain is indicated. (C) Surface representation of the composite cryo-EM density map of human RNF213. (D) Cartoon representation of the human RNF213 structure, rotated 180° relative to Fig. 3C. Colours match domains in Fig. 3B. (E) CBM20 domain of RNF213 as predicted by AlphaFold. (F) CBM20 of glucoamylase from *Aspergillus niger* (PDB 1AC0). Sugars shown as sticks. (G) Cartoon representation of human RNF213 with docked AlphaFold prediction. Positively selected residues are highlighted as red spheres. (H) Percentage of cytosolic *S. Typhimurium* positive for GFP-RNF213 at 4 h post-infection in RNF213$^{KO}$ MEFs stably expressing the specified GFP-RNF213 alleles. Mean +/− SEM of $n = 3$ independent biological experiments, each performed in technical triplicates. $n > 100$ bacteria per coverslip. One-way ANOVA test, *$p = 8.3 \times 10^{-5}$. (I) Percentage of *L. monocytogenes ΔActA* colonies positive for GFP-RNF213 at 6 h post-infection in RNF213$^{KO}$ MEFs stably expressing the specified GFP-RNF213 alleles. Mean +/− SEM of $n = 3$ independent biological experiments, each performed in technical triplicates. $n > 100$ colonies per experiment. One-way ANOVA test, *a $p = 0.0042$, *b $p = 4.8 \times 10^{-7}$. (J) Percentage of *T. gondii* Type I RH vacuoles positive for GFP-RNF213 at 1 h post-infection in RNF213$^{KO}$ MEFs stably expressing the specified GFP-RNF213 alleles. Mean +/− SEM of $n = 16$ positions automatically acquired and counted per triplicate wells in $n = 4$ independent biological experiments. One-way ANOVA test, *a $p = 6.1 \times 10^{-6}$, *b $p = 7.4 \times 10^{-6}$. (K) Confocal micrographs of RNF213$^{KO}$ MEFs stably expressing the indicated GFP-tagged RNF213 variants. Scale bar, 20 μm. Source data are available online for this figure.

metal ion, presumably $Zn^{2+}$, which is coordinated by four evolutionarily conserved Cys and His residues (C4505, H4509, C4525, C4528). In contrast, C4516 and H4537 are not resolved in the cryo-EM structure. The AlphaFold model predicts that residues C4516 and H4537 are located on the opposite end of the domain near each other, consistent with their roles as nucleophile and general base in the E2–E3 transthiolation and the E3-substrate transfer reactions, respectively. To test the contribution of the RZ finger to pathogen and vacuole ubiquitylation, we generated mutant alleles of RNF213 predicted to disrupt either metal binding (H4509A) or catalytic activity of the RZ finger (C4516S).

To test for the role of the RING domain and the RZ finger, we complemented RNF213$^{KO}$ MEFs with the indicated GFP-RNF213 alleles, followed by infection with *S. Typhimurium*, *L. monocytogenes ΔactA* or the RH strain of *T. gondii* (Figs. 4D–F and EV4). Our analyses revealed that RNF213 alleles with defects in the RZ domain (RNF213$_{H4509A}$ or RNF213$_{C4516S}$) failed to ubiquitylate *S. Typhimurium*, *L. monocytogenes ΔactA*, or the *T. gondii* vacuole. In contrast, inactivation of the RING domain in RNF213$_{ΔRING}$ or RNF213$_{H4014N}$ did not affect the ubiquitylation of *S. Typhimurium* and *L. monocytogenes* and only partially impaired ubiquitylation of the *T. gondii* vacuole. Alleles targeting the RZ finger caused defects in both the ubiquitylation of and RNF213 coat formation on *S. Typhimurium* and *L. monocytogenes ΔactA*. In contrast, RNF213 coats on *T. gondii* vacuoles were still formed in RZ finger mutants (RNF213$_{H4509A}$ and RNF213$_{C4516S}$) despite lack of ubiquitin deposition. We conclude that the RZ finger-mediated ubiquitylation is essential for RNF213 coat formation on bacterial surfaces but not on the *T. gondii* vacuole.

## Discussion

The E3 ligase RNF213 has emerged as a restriction factor against a surprisingly wide range of phylogenetically distant intracellular pathogens, including the eukaryotic parasite *T. gondii*, the Gram-negative bacterium *S. Typhimurium*, the Gram-positive bacterium *L. monocytogenes*, as well as RNA and DNA viruses (Otten et al, 2021; Thery et al, 2021; Hernandez et al, 2022; Walsh et al, 2022; Houzelstein et al, 2021; Matta et al, 2023). How RNF213 senses

such diverse targets is a question of fundamental importance to immunology.

By solving the cryo-EM structure of human RNF213, we discovered a carbohydrate-binding CBM20 domain in its previously unresolved N-terminus. We also found that simian RNF213 genes are under intense positive selection, particularly in their CBM20 domain and adjacent parts of the stalk region. Our data therefore indicate that an evolutionary arms race for direct binding between RNF213 and its microbial target(s) has shaped RNF213 function, consistent with RNF213 serving as a pattern recognition receptor that directly interacts with pathogen-derived ligand(s). Given the clustering of positively selected residues to the CBM20 domain (and the adjacent stalk) and the prominent role of microbial carbohydrates in immune recognition, we speculate that the CBM20 domain may serve as a critical receptor domain in RNF213.

We found that RNF213 forms dense protein coats both on the surface of *S. Typhimurium* and *L. monocytogenes* that have escaped from their vacuoles into the cytosol as well as on the membrane of parasitophorous vacuoles surrounding *T. gondii*. Coat formation is not controlled by diffusion-limited recruitment of RNF213 from the cytosol. Rather, coat formation involves rare, rate-limiting initiation events, followed by rapid multi-directional growth of the coat that is reliant on cooperative incorporation of further RNF213 molecules. Assuming *T. gondii* parasitophorous vacuoles represent a surface area of 30 μm² that becomes coated with a monolayer of tightly packed RNF213 molecules, each with a footprint of 100 nm², we estimate that during coat expansion (5.0 min) around 1000 RNF213 molecules per second are added to the RNF213 coat. The rate of RNF213 incorporation during coat expansion (1000 molecules/s) is therefore approximately one million-fold faster than during coat initiation (1 molecule/27 min) assuming synchronous parasite invasion. This estimate remains true even if the formation of the RNF213 coat on all tested pathogens requires catalytic activity in the hexameric AAA+ ATPase ring of RNF213. In contrast, E3 ligase activity mediated by the RZ finger is required for coat formation on *S. Typhimurium* and *L. monocytogenes* but not on the parasitophorous vacuoles surrounding *T. gondii*. We conclude that RNF213 responds to evolutionarily distant pathogens through an enzymatically amplified cooperative recruitment mechanism that results in dense RNF213 coats on target structures.

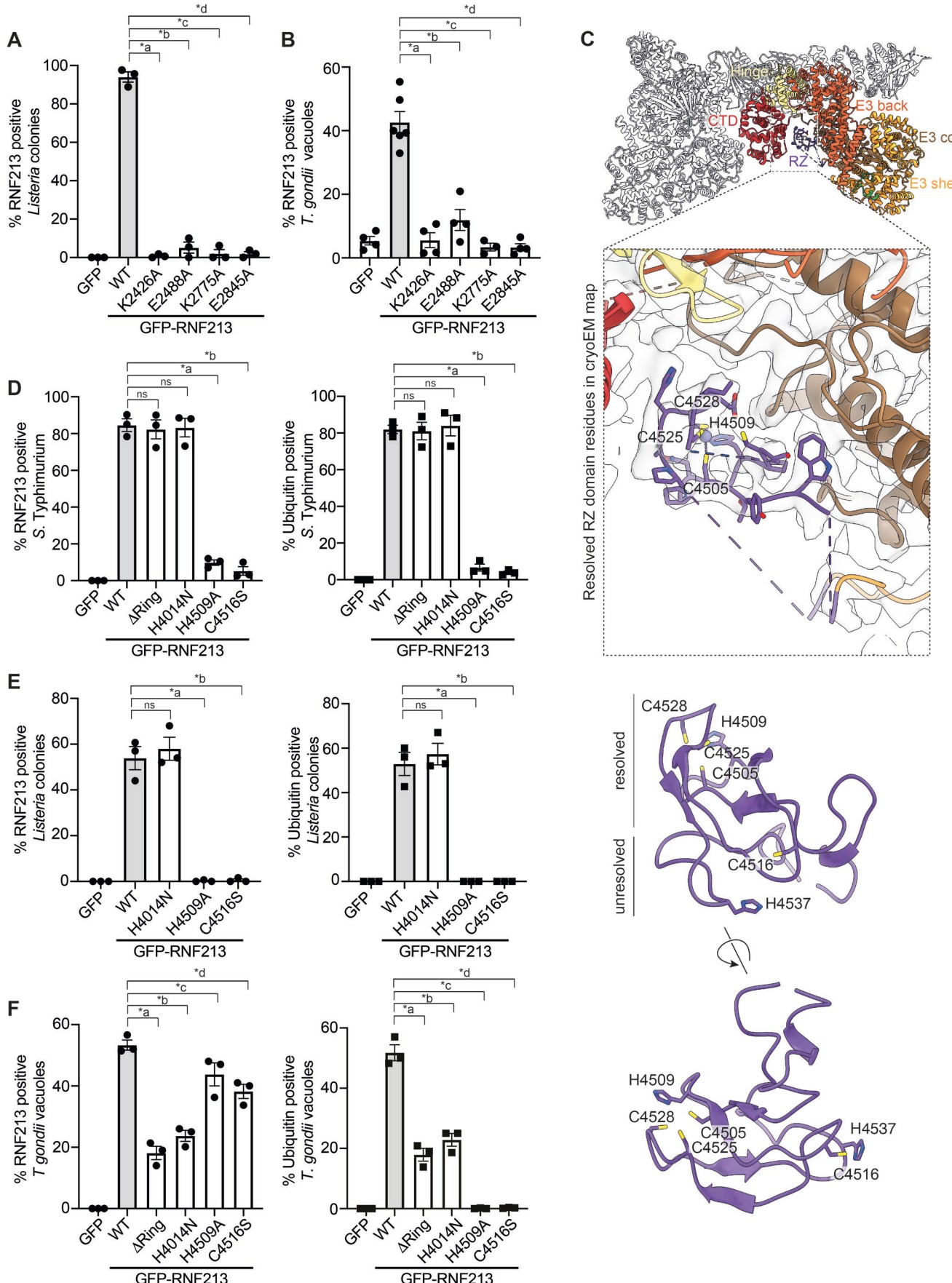

**Figure 4. Enzymatic activity in RNF213 is required for coat formation on pathogens.**

(A, B) RNF213$^{KO}$ MEFs expressing the indicated GFP-RNF213 alleles. (A) Percentage of *L. monocytogenes* Δ*ActA* colonies positive for GFP-RNF213 at 6 h post-infection. Mean $+/-$ SEM of $n = 3$ independent biological experiments, each performed in technical triplicates. $n > 100$ colonies per experiment. One-way ANOVA test (*a $p = 8 \times 10^{-13}$; *b $p = 1.4 \times 10^{-12}$; *c $p = 9 \times 10^{-13}$; *d $p = 9 \times 10^{-13}$). (B) Percentage of *T. gondii* Type I RH vacuoles positive for GFP-RNF213 at 1 h post-infection. Mean $+/-$ SEM of $n = 16$ positions, automatically acquired and counted, from triplicate wells in $n = 4$ independent biological experiments. One-way ANOVA test (*a $p = 1.6 \times 10^{-7}$; *b $p = 2 \times 10^{-6}$; *c $p = 2.6 \times 10^{-7}$; *d $p = 7.3 \times 10^{-8}$). (C) Structure of the RZ domain. Top panel: Cartoon representation of the structure of human RNF213, determined by cryo-EM. RZ domain, E3 module, hinge domain and CTD highlighted in different colours. Insert: Cryo-EM density and partly resolved RZ domain. Dotted lines link the resolved part of the RZ domain to the C-lobe of the E3 shell domain. Middle and bottom panel: AlphaFold prediction of the RZ domain, positioned as in the insert (middle panel) and rotated (bottom panel). Note the positioning of H4509, C4505, C4525 and C4528, predicted to form a metal ion binding site, and of C4516 and H4537, predicted to act as nucleophile and general base in the E2-E3 transthiolation and E3-substrate ubiquitin transfer reactions, respectively. (D–F) RNF213$^{KO}$ MEFs expressing the indicated GFP-RNF213 alleles. (D) Percentage of cytosolic *S. Typhimurium* positive for GFP-RNF213 and ubiquitin (FK2) at 4 h post-infection. Mean $+/-$ SEM of $n = 3$ independent biological experiments, each performed in technical triplicates. $n > 100$ bacteria per coverslip. One-way ANOVA test (*a $p = 1.5 \times 10^{-6}$; *b $p = 1.0 \times 10^{-6}$). (E) Percentage of *L. monocytogenes* Δ*ActA* colonies positive for GFP-RNF213 and ubiquitin (FK2) at 6 h post-infection. Mean $+/-$ SEM of $n = 3$ independent biological experiments, each performed in technical triplicates. $n > 100$ colonies per experiment. One-way ANOVA test (*a $p = 0.000024$; *b $p = 0.000025$). (F) Percentage of *T. gondii* Type I RH vacuoles positive for GFP-RNF213 and ubiquitin (FK2) at 1 h post-infection. Mean $+/-$ SD of $n = 3$ independent biological experiments, each performed in technical triplicates. $n > 200$ parasites per coverslip. One-way ANOVA test, RNF213 positive *T. gondii* vacuoles; *a $p = 2.5 \times 10^{-6}$; *b $p = 1.2 \times 10^{-5}$; *c $p = 0.025$; *d $p = 0.0029$. Ubiquitin positive *T. gondii* vacuoles *a $p = 3 \times 10^{-7}$; *b $p = 1.7 \times 10^{-6}$; *c $p = 6.5 \times 10^{-9}$; *d $p = 6.7 \times 10^{-9}$. Source data are available online for this figure.

The mechanism of RNF213 recruitment to intracellular pathogens is most unusual for a pattern recognition receptor and deserves further consideration. Coat formation through distinct initiation events rather than a diffusion-limited and mass action-controlled process of RNF213 recruitment suggests that ligand and/or receptor are not freely available to interact. For example, ligand(s) could be hidden inside the extensive O-antigen layer covering *S. Typhimurium* and may become accessible only briefly, for instance when temporary gaps form in the O-antigen layer. RNF213 AAA+ ATPase activity could help provide access to concealed ligands, enabling RNF213 to penetrate the O-antigen layer of *S. Typhimurium* in order to access the innermost lipid A moiety of LPS, the minimal substrate of LPS ubiquitylation by RNF213 (Otten et al, 2021). Thus, AAA+ ATPase activity-driven access to lipid A would occur along tracks of repetitive O-antigen subunits, resembling movement of dynein along microtubules. This model has additional interesting parallels to the cytosolic LPS receptor Caspase 4, which also relies on nucleotide hydrolysis to access LPS embedded in bacterial membranes, with GTP hydrolysis provided in trans by guanylate-binding proteins (GBPs) (Wandel et al, 2020; Fisch et al, 2019; Pilla et al, 2014; Santos et al, 2018, 2020). Thus, nucleotide hydrolysis may represent a common theme for cytosolic receptors attempting to access lipid A embedded in bacterial membranes. Even though the identity of RNF213 ligands on *L. monocytogenes* and the parasitophorous vacuoles containing *T. gondii* is still unknown, based on the unusual coat formation, these ligands may be similarly difficult to access as the lipid A moiety of LPS on *S. Typhimurium*. An alternative explanation for the unusual mode of coat formation is that RNF213 may predominantly exist in an auto-inhibited state and that the protein's intrinsic AAA+ ATPase activity is required to relieve this autoinhibition.

*RNF213* encodes two potential ubiquitin ligase domains, a canonical RING domain and the recently identified RZ finger, an unusual feature among the large family of ubiquitin E3 ligases. The RZ finger was originally defined based on homology between RNF213 and ZNFX1 (Otten et al, 2021), with the occurrence of four conserved cysteines and two conserved histidines prompting us to predict a Zn$^{2+}$ coordinating structure. The cryo-EM structure of human RNF213 and the Alphafold models of the RZ finger reported here and by others (Ahel et al, 2021) fully support our prediction, with three cysteines and one histidine (C4505, H4509, C4525, C4528) arranged to coordinate a metal ion and Cys4516 and His4537 located near each other on the opposite end of the domain. Based on these structures, we predict that Cys4516 acts as the nucleophile accepting ubiquitin from the E2 in a transthiolation reaction, while His4537 acts as a general base in the subsequent transfer of ubiquitin onto its ultimate substrate, similar to a canonical cysteine/histidine pair in RBR ligases (Stieglitz et al, 2013; Walden and Rittinger, 2018). Consistent with our model, we find that mutations in the RZ domain of RNF213 predicted to either impair Zn$^{2+}$ coordination (H4509A) or to prevent ubiquitin transfer onto RNF213 (C4516S) failed to ubiquitylate any of the three substrates tested: *S. Typhimurium*, *L. monocytogenes*, or parasitophorous vacuoles containing *T. gondii*. In contrast, deletion of the RING domain had no effect on pathogen ubiquitylation, as reported previously for *S. Typhimurium* and for autoubiquitylation (Otten et al, 2021; Ahel et al, 2020). Thus, although the isolated RING domain is active in vitro (Liu et al, 2011; Takeda et al, 2020; Bhardwaj et al, 2023; Habu and Harada, 2021), the activity of RNF213 in vivo relies on its RZ finger. No bona fide substrate for the RING domain has been identified so far. Moreover, how ubiquitin deposition promotes formation of RNF213 protein coats remains unclear. One possibility is that incoming RNF213 monomers oligomerize with RNF213 molecules already present in the coat, in addition to binding cognate PAMP(s), thus leading to enhanced affinity and coincidence detection.

Our findings reveal that RNF213 coat formation is a surprisingly universal process across diverse pathogens and pathogen-encasing host membranes. Tight regulation of RNF213 coat formation may be needed to prevent uncontrolled activation of RNF213 and avoid pathological side effects as RNF213 affects many cellular processes including fatty acid metabolism, lipid droplet homoeostasis, non-mitochondrial oxygen consumption, and NF-κB signalling (Piccolis et al, 2019; Sugihara et al, 2019; Banh et al, 2016; Takeda et al, 2020). Although positive selection has acted mainly on the N-terminus of RNF213, mutations in patients with Moyamoya disease cluster towards the C-terminus and are primarily found in the E3 module. While the precise function of RNF213 in Moyamoya disease remains to be identified, disturbances in substrate-induced oligomerization of RNF213 due to mutations in its E3 module may contribute to Moyamoya aetiology.

# Methods

**Reagents and tools table**

| Reagent/Resource | Reference or Source | Identifier or Catalog Number |
| --- | --- | --- |
| **Experimental Models** | | |
| HeLa cells (H. sapiens) | | RRID: CVCL_0030 |
| MEFs (M. musculus) | Dr Chihiro Sasakawa | |
| Type I-Tomato (RH) (T. gondii) | Dr Eva-Maria Frickel | |
| Type I-Tomato (Pru) (T. gondii) | Dr Eva-Maria Frickel | |
| Listeria monocytogenes ΔActA expressing RFP | Dr John Brumell | Strain 1043S |
| Salmonella enterica serovar Typhimurium, strain 12023 | Dr David Holden | Strain 12023 |
| **Recombinant DNA** | | |
| GO244_PB-CAG-GIP-rtTA3-TRE-FlagGFP-RNF213 | Otten et al (2021)(Nature) | |
| GO273_OP806_pACEBac1 2xStrep-RNF213-3xFLAG | Otten et al (2021)(Nature) | |
| GO253_PB-CAG-GIP-rtTA3-TRE-FlagGFP-RNF213 368 EVKA - end | Otten et al (2021)(Nature) | |
| GO254_PB-CAG-GIP-rtTA3-TRE-FlagGFP-RNF213 585 VKRY - end | Otten et al (2021)(Nature) | |
| AC10_PB CAG GIP rtTA3 TRE tight GFP_designed | Otten et al (2021)(Nature) | |
| AC8_ pCMV_HyPBase | Otten et al (2021)(Nature) | |
| GO245_PB-CAG-GIP-rtTA3-TRE-FlagGFP-RNF213 K2426A | Otten et al (2021)(Nature) | |
| GO246_PB-CAG-GIP-rtTA3-TRE-FlagGFP-RNF213 E2488A | Otten et al (2021)(Nature) | |
| GO247_PB-CAG-GIP-rtTA3-TRE-FlagGFP-RNF213 K2775A | Otten et al (2021)(Nature) | |
| GO248_PB-CAG-GIP-rtTA3-TRE-FlagGFP-RNF213 E2845A | Otten et al (2021)(Nature) | |
| GO257_PB-CAG-GIP-rtTA3-TRE-FlagGFP-RNF213 dRING | Otten et al (2021)(Nature) | |
| GO258_PB-CAG-GIP-rtTA3-TRE-FlagGFP-RNF213 H4014N | Otten et al (2021)(Nature) | |
| GO264_PB-CAG-GIP-rtTA3-TRE-FlagGFP-RNF213 H4509A | Otten et al (2021)(Nature) | |
| CH2_ PB-CAG-GIP-rtTA3-TRE-FlagGFP-RNF213 C4516S | This study | |
| AC89_ PB-CAG-GIP-rTTA3-TRE-FlagGFP-RNF213 int_del 378-515, CBM | This study | |
| AC89_ PB-CAG-GIP-rTTA3-TRE-FlagGFP-RNF213 int_del 378-503, CBM | This study | |
| MY053_ PiggyBac-Flag-GFP-RNF213 (F391G/W413/Y462) | This study | |
| MY059_ PiggyBac-Flag-GFP-RNF213 (H421G/W413/Y462) | This study | |
| MY061_ PiggyBac-Flag-GFP-RNF213 (F401G/W413/Y462) | This study | |

| Reagent/Resource | Reference or Source | Identifier or Catalog Number |
|---|---|---|
| MY063_ PiggyBac-Flag-GFP-RNF213 (Y422G/W413/Y462) | This study | |
| **Antibodies** | | |
| FK2 | Enzo Life Science | BML-PW8810 |
| α-RNF213 | Sigma | HPA 003347 |
| Donkey anti-rabbit 488 | Invitrogen | A21206 |
| Donkey anti-mouse 647 | Invitrogen | A31571 |
| Donkey anti-mouse 488 | Invitrogen | A21202 |
| **Oligonucleotides and other sequence-based reagents** | | |
| **Primer ID** | | **Sequence** |
| AC_154 CBMdel_378-515 For | This study | CTCCAGGCGGCGACGGCCCCAGAA |
| AC_155 CBMdel_378-515 Rev | This study | TTCTGGGGCCGTCGCCGCCTGGAG |
| AC_156 CBMdel_378-503 For | This study | GTCTCCAGGCGGCCCCCACGGCAGACTC |
| AC_157 CBMdel_378-503 Rev | This study | GAGTCTGCCGTGGGGGCCGCCTGGAGAC |
| VD_13 RNF213Y462A Rev | This study | GTAGATGAACTCGgcCTCGAAGGACTCG |
| VD_14 RNF213Y462A For | This study | CGAGTCCTTCGAGgcCGAGTTCATCTAC |
| VD_15 RNF213W413A For | This study | GGCGAGAGCAAGgcGGACAGCAATAT |
| VD_16 RNF213W413A Rev | This study | ATATTGCTGTCCgcCTTGCTCTCGCC |
| CJ_85 RNF213 For | This study | ATGGACGAGCTGTACAAGgAC |
| CJ_86 391/393 Rev | This study | GTGCAGGCTGATAATAGCGTG |
| CJ_87 F391G For | This study | ACGCTATTATCAGCCTGCACggcCCCTTCAATCCCGACCTGCAC |
| CJ_99 RNF-BglII Rev | This study | TCCTCGGTCACATTGGCCAAG |
| CJ_89 398/401 Rev | This study | CAGGTCGGGATTGAAGGGGAAG |
| CJ_91 F401G For | This study | TCCCCTTCAATCCCGACCTGCACAAGGTGggcATCAGAGGCGGCGAAGAGTTC |
| CJ_92 421/422 For | This study | CAGCTCGCAGATATTGCTGTC |
| CJ_93 H421G For | This study | ACAGCAATATCTGCGAGCTGggcTACACCCGCGACCTGGGACATG |
| CJ_94 Y422- For | This study | ACAGCAATATCTGCGAGCTGCACggcACCCGCGACCTGGGACATGATAG |
| CH_2 GO443 For | This study | tccagaggaagagagccggccaacgAGGCCTCCGTGGAATACC |
| None GO589 Rev | This study | TCTGTTCCATTGGTCTGCCAgaCTCTCC |
| CH_3 GO590 For | This study | ACCCTTGCAGCGTGGGAGAGtcTGGCAGACCAATGGAACAGAGCA |
| None CMH5 Rev | This study | GCTCTCGTTAATTAACTCGACTAGGC |
| **Chemicals, Enzymes and other reagents** | | |
| PEI | Polysciences | Cat # 24765 |
| Universal nuclease | Pierce | Cat #88700 |
| Benzonase nuclease | Sigma-Aldrich | Cat #9025-65-4 |
| EDTA-free protease inhibitor tablets | Roche | Cat #11836170001 |
| HiFi | New England Biolabs | Cat # E2621L |
| StrepTrap HP column | Cytiva | Cat # 28-9075-46 |
| IMDM | Gibco | Cat #31980-022 |
| FCS | Gibco | |
| Lipofectamine 2000 | Invitrogen | Cat #11668-027 |
| Puromycin | Gibo | Cat # A11138-03 |
| ZVAD | | Cat # G7231 |
| Doxycycline | Toku-E | Cat # 24390-14-5 |
| 24-well plates | Corning | Cat # 3526 |

| Reagent/Resource | Reference or Source | Identifier or Catalog Number |
|---|---|---|
| DMEM | Gibo | Cat # 31966-021 |
| Gentamycin | Gibo | Cat # 15750-045 |
| Glass bottom Falcon 96-well plate | Falcon | Cat # 353219 |
| Glass bottom Ibidi 24-well plate | Ibidi | Cat # 82426 |
| BSA | Sigma-Aldrich | Cat # A7638-5G |
| 35 mm glass bottom mattek microwell dishes | Mattek | Cat # P35G-1.5-14-C |
| Leibovitz's L-15 media | Gibco | Cat # 11415064 |
| **Software** | | |
| RELION-5.0 | https://relion.readthedocs.io/en/release-5.0/ | |
| cryoSPARC-4.2 | https://cryosparc.com | |
| CTFFIND-4.1 | https://grigorieflab.umassmed.edu/ctffind4 | |
| EPU | ThermoFisher Scientific | https://www.thermofisher.com/uk/en/home/electron-microscopy/products/software-em-3d-vis/epu-software.html |
| Coot | https://www2.mrc-lmb.cam.ac.uk/personal/pemsley/coot/ | |
| PHENIX | https://phenix-online.org | |
| Chimera X | https://www.cgl.ucsf.edu/chimerax/ | |
| FiJi ImageJ | NIH | https://imagej.nih.gov/ij/ |
| Nikon NIS elements software | Nikon | https://www.microscope.healthcare.nikon.com/products/software |
| Prism | Graphpad | https://www.graphpad.com/scientific-software/prism/ |
| PHYML | http://www.atgc-montpellier.fr | PhyML : "A simple, fast, and accurate algorithm to estimate large phylogenies by maximum likelihood." Guindon S, 2003 |
| Python | Python Software Foundation | https://www.python.org |
| **Other** | | |
| Titan Krios | ThermoFisher Scientific | |
| Microtip sonicator | Vibra Cell | |
| Zeiss 780 inverted microscope | Nikon | |
| Nikon HCA microscope | Nikon | |
| Zeiss Axio Imager | Zeiss | |
| Nikon iSIM swept field microscope | Nikon | |
| Nikon X1 spinning disk | Nikon | |

## Plasmids

To express RNF213 in mammalian cells an inducible PiggyBac transposon system was used (Otten et al, 2021). Human RNF213 alleles were generated by PCR from a codon-optimized RNF213 cDNA (Otten et al, 2021). Constructs were verified by sequencing.

## Protein expression and purification

Wild-type human RNF213 was expressed in insect cells and purified as previously described (Otten et al, 2021). Briefly, pOP806_pACEBac1 2xStrep-RNF213-3xFLAG plasmid was transformed into DH10EmBacY cells (DH10Bac with YFP reporter).

Blue–white screening was used to isolate colonies containing recombinant baculoviral shuttle vectors (bacmids) and bacmid DNA was extracted combining cell lysis and neutralization using buffer P1, P2 and N3 (Qiagen), followed by isopropanol precipitation. A 6-well plate of *Spodoptera frugiperda* (Sf9) cells (Oxford Expression Technologies) grown at 27 °C in Insect-Xpress (Lonza) without shaking was transfected with bacmid plasmid using PEI transfection reagent. After 6 days, virus P1 was collected and used 1:25 to transduce 50 mL ($1.8 \times 10^6$ cells/ml) of Sf9 cells. After 7 days of incubation at 27 °C with 140 rpm shaking, virus P2 was collected. To express protein, 1 L ($2.6 \times 10^6$ cells/mL) of Sf9 cells was transduced with 1:50 dilution of P2 virus and incubated at 27 °C with 140 rpm shaking for 72 h. Cells were pelleted by

centrifugation, snap-frozen in liquid nitrogen, and stored at −80 °C. To lyse cells, the pellets were thawed and resuspended to 180 mL total volume in lysis buffer (30 mM HEPES, 100 mM NaCl, 10 mM $MgCl_2$, 0.5 mM TCEP, pH 7.6), containing 20 µL universal nuclease (Pierce), 20 µL benzonase nuclease (Sigma-Aldrich), and 2 EDTA-free protease inhibitor tablets (Roche). The cell suspension was stirred for 1 h and then sonicated for 50 s in 5-s pulses with 25-s waiting time at 70% amplitude using a 130 W microtip sonicator (Vibra Cell). The lysate was centrifuged at $20,000 \times g$ for 60 min at 4 °C. The clarified lysate was filtered through a 0.2 µm filter (Millipore) and applied to $3 \times 5$-mL StrepTrap HP columns (GE Healthcare), connected in series. After washing the columns with 100 mL lysis buffer, RNF213 was eluted with lysis buffer supplemented with 2.5 mM desthiobiotin, pH 8. The eluted protein was kept at 4 °C and used immediately for cryo-EM specimen preparation. All purification steps were carried out at 4 °C, using an ÄKTA pure 25 (GE Healthcare).

## Cryo-EM specimen preparation

Purified human RNF213 was used at the peak elution concentration of 5.0 mg/mL in the elution buffer (30 mM HEPES, 100 mM NaCl, 10 mM $MgCl_2$, 0.5 mM TCEP, 2.5 mM desthiobiotin, pH 8) to prepare the cryo-EM specimen. An all-gold UltrAuFoil R 0.6/1 300-mesh grid (Russo and Passmore, 2014) was pre-cleaned using 9:1 $Ar:O_2$ plasma in a Fischione plasma chamber for 180 s at 100% power, 30 sccm gas flow. A manual plunger of the Talmon type (Bellare et al, 1988), situated in a 4 °C cold room was used to vitrify the specimen. Three microlitres of the specimen were applied to the foil side of the grid and blotted from the same side for 14 s using Whatman No 1 filter paper. The grid was then immediately plunged frozen into liquid ethane, held at 93 Kelvin in a temperature-controlled cryostat (Russo et al, 2016).

## Cryo-EM data collection

Electron micrographs were collected during a 24-h data collection session on a Titan Krios (Thermo Fisher Scientific) electron microscope at the UK national electron Bio-Imaging Centre (eBIC). The microscope was operated at 300 keV and micrographs were acquired on a Falcon 4i direct electron detector after energy filtering through a 10 eV slit using a Selectris X imaging filter. The nominal magnification was 130,000×, corresponding to magnified pixel size of 0.921 Å. The beam diameter was set to 0.9 µm, arranging for an electron flux of 5.2 e⁻/pix/s, which, during the total exposure time of 4.86 s, corresponds to total electron fluence of 29.8 e⁻/Å². Automated data collection of one exposure per hole was performed with aberration-free image shift using the EPU software. All 7022 multi-frame electron micrographs were saved in EER format. The data collection settings are summarized in Appendix Table S2.

## Cryo-EM data processing

The electron micrographs were processed using a combination of RELION-5.0 and cryoSPARC-4.2 (Punjani et al, 2017; Kimanius et al, 2021). The data processing is summarized in Fig. EV3C–F. Briefly, all cryo-EM movies were imported in RELION and motion corrected after grouping the EER frames into 30 fractions, each

corresponding to 1.0 e⁻/Å² of irradiation and applying gain correction. The contrast transfer functions were estimated using CTFFIND-4.1 (Rohou and Grigorieff, 2015) Initial particle picking was performed using the Laplacian-of-Gaussian (LoG) picker in RELION with diameter range from 200 Å to 350 Å and minimal threshold at −1 standard deviations. The picked 1,010,981 particles were extracted into 512-pixel boxes and binned by a factor of 2 to 256 pixels at 1.842 Å/pix. The extracted particles were imported into cryoSPARC for 2D and 3D classification, where false picks and damaged particles were removed, reducing the total particle number to 143,490. These particles were imported back into RELION, re-extracted at the original pixel size, and subjected to multiple rounds of optical aberration refinement (Zivanov et al, 2020), Bayesian polishing (Zivanov et al, 2019), and 3D refinement with regularization by the Blush algorithm (Kimanius et al, 2024). Finally, masks were created to split RNF213 into five regions which are somewhat flexible relative to each other, as indicated by blurring of the composite map and by the molecular motions estimated using DynaMight (Schwab et al, 2024). Each region was independently refined with a local search around the refined angles and offsets and regularization by Blush. The composite map of these five regions (CBM, stalk, ATPase, and two regions in the E3) was used for model building (Fig. EV3C).

## Atomic model building and refinement

The initial model was constructed using an AlphaFold 5 prediction for full-length human RNF213 and docked in the composite map (Jumper et al, 2021). The model was initially refined into the composite map using tight Gemman-McClure restraints in Coot (Emsley et al, 2010), followed by real-space refinement in PHENIX (Afonine et al, 2018), and final manual adjustments in Coot. Regions which are not present on the map were removed from the model after docking. An exception was made for the carbohydrate binding module (CBM) for completeness, where the whole fragment was kept in the atomic model, as predicted by AlphaFold, despite some parts of it not being well resolved in the map (Fig. EV3).

## Cell culture and Piggybac transfection

*Mycoplasma*-negative wild type mouse embryonic fibroblasts (MEFs) kindly provided by Dr Chihiro Sasakawa and the derivative RNF213[KO] MEFs[3] were grown in IMDM supplemented with 10% FCS at 37 °C in 5% $CO_2$. An inducible Piggybac transposon system was used to generate RNF213[KO] stably expressing different RNF213 alleles, as described in (Otten et al, 2021). Briefly, RNF213[KO] MEFs seeded in 24-well plates were transfected with 1 µg of PiggyBac plasmid (Glover et al, 2013) and 1 µg of pBase (Yusa et al, 2011) using Lipofectamine 2000 and 2 days later cells were selected with puromycin. Protein expression was induced with 1 µg/mL doxycycline for at least 15 h in the presence or absence of 20 µM ZVAD.

## Biosafety

All experiments involving live *Toxoplasma gondii*, *Listeria monocytogenes* and *Salmonella enterica* serovar Typhimurium were performed at the MRC Laboratory of Molecular Biology in dedicated Containment Level 2 laboratories. Biological Risk

Assessments and Standard Operating Procedures were approved by the LMB Biological Safety Committee and, where appropriate, notified to the UK Health and Safety Executive.

## *Toxoplasma gondii* strains and passaging

Type I-Tomato (RH) and Type II-Tomato (Pru) were kindly provided by Eva-Maria Frickel. The parasites were passaged in monolayers of MEFs cells and tachyzoites were harvested upon eggression from the cells. For infections, eggressed parasites were added at 1:20 dilution onto cells, span for 1 min at 1000 rpm and let to infect for 1 h at 37 °C. Then cells were washed with warm PBS and fixed with 4% paraformaldehyde for 15 min.

## Bacterial infections

*Listeria monocytogenes ΔActA* expressing red fluorescent protein (PL1940, actAp-tagRFP at tRNAArg in 10403S (WT) was kindly gifted by John Brumell. A single colony of *L.monocytogenes* was grown until saturation in 2 mL of brain heart broth at 30 °C and sub-cultured (1:10) in fresh brain heart broth until they reached $OD_{600} = 0.8$–0.9 at 37 °C and shaking before infection. Bacteria was then washed 3 times in warm PBS (by spinning $10,000 \times g$ for 2 min) and resuspended in warm DMDM. For the counting of recruitment of different RNF213 alleles and ubiquitylation, MEFs in 24-well plates growing in IMDM supplemented with 10% FCS received 60 µL of bacteria suspension. They were span for 1 min at 1000 rpm and let to infect for 1 h at 37 °C. Infections were then washed 3 times with warm PBS and incubated for 5 h in IMDM supplemented with 10% FCS and 20 µg/ml gentamycin. Next, infections were stopped by washing twice in warm PBS and fixing with 4% paraformaldehyde for 15 min.

A single colony of *Salmonella* Typhimurium expressing red fluorescent protein was grown until saturation in 1 mL of LB at 37 °C and 700 rpm. It was sub-cultured (1:33) in fresh LB for 3.5 h at 37 °C and shaking 700 rpm before infection. For the counting of recruitment of different RNF213 alleles and ubiquitylation, MEFs in 24-well plates growing in IMDM supplemented with 10% FCS received 20 µL of 1:15 diluted bacteria suspension for 8 min at 37 °C. Infections were then washed twice with warm PBS and incubated for 3 h in IMDM supplemented with 10% FCS and 100 µg/ml gentamycin. Next, infections were stopped by washing twice in warm PBS and fixing with 4% paraformaldehyde for 15 min.

## Microscopy of fixed cells

Cells were grown on either glass coverslips, glass bottom Falcon 96-well plate or glass bottom Ibidi 24-well plates. After infection and fixation in 4% paraformaldehyde, cells were washed twice in PBS, permeabilised for 5 min in PBS with 0.1% Triton X-100 and blocked with PBS with 2% BSA for 1 h. Cells were incubated overnight with primary antibodies followed by Alexa conjugated secondary antibodies in blocking solution for at least 1 h at room temperature. Coverslips were then mounted in Prolong gold mounting medium. Images of infections were acquired from coverslips on a Zeiss 780 inverted microscope using a 63x/1.4NA Oil immersion lens. Images were analysed using FiJi ImageJ. Cells growing in glass bottom plates were imaged on a Nikon HCA

microscope using a 20x/0.75NA Air lens, acquiring at least 24 positions per condition that were automatically analysed and quantified using the Nikon NIS Elements software.

Marker-positive bacteria were scored by eye from coverslips on a Zeiss Axio Imager microscope using a 100x/1.4NA Oil immersion lens, amongst at least 200 bacteria per coverslip. At least three biological repeats with two technical replicates each were performed. Bacteria were scored by visual counting of $n > 200$ bacteria per replicate. In the case of *Salmonella* Typhimurium, only large (cytosolic) bacteria were considered for quantification. Graphs show mean ± S.E.M.

## Live-cell microscopy and measurement of RNF213 coat formation

Cells seeded in 35 mm glass bottom MatTek microwell dishes were infected with *S.* Typhimurium, *L. monocytogenes* or *T. gondii* as described above for 20 min, 60 min or 5 min, respectively, washed four times in warm PBS and incubated in Leibovitz's L-15 supplemented with 100 µg/ml gentamycin, 20 µg/ml gentamycin or no gentamycin (for *Salmonella*, *Listeria* or *Toxoplasma* infections, respectively) for the duration of the experiment. A Nikon iSIM swept field high speed inverted microscope with a 100X super resolution Apo TIRF oil objective or a Nikon X1 Spinning Disk inverted microscope with a 40x/1.3NA Oil lens were used to acquire images.

A custom macro for Fiji (Schindelin et al, 2012) was written to measure the recovery speed of the fluorescently labelled coat around motile parasites in time lapse imaging experiments. To simplify the downstream analysis, regions of interest (i.e., parasites) were identified manually in the 3D image sequences. To obtain 3D masks corresponding to selected parasites over time, a median filter and threshold were applied to the channel with the parasite marker. The centermost label was then tracked over time looking for the nearest centroid at subsequent time points and a 3D watershed (Legland et al, 2016) was used to separate temporarily touching parasites. A 1um shell around the parasite masks was computed and the mean intensity inside the shell was measured. The intensity profiles were exported as csv files and adjusted to an exponential recovery model

$$f(t) = a + b\left(0.5 + 0.5 \cdot \text{erf}\left(\frac{x - c}{d}\right)\right) \cdot e^{-et}$$

to the intensity profile over time. A 5% to 95% completion time was then computed from the parameters of the model as $2.38 \cdot d$.

The initiation times of RNF213 coats on *T. gondii* parasitophorous vacuoles ($n = 27$ data points in total) were calculated as the time points of 5% of the maximal GFP intensity from the per-parasite coating curve fits (Fig. EV2). The probability P (T_initiation >t) that coat initiation starts later than some time t was estimated as the fraction of observed parasites whose coat initiated later than t amongst all 27 parasites used in this analysis, and is plotted as a function of time with the '+' symbols in Fig. 2F. Assuming that coat initiation follows the probability distribution for the time of first arrival of an event obeying Poisson statistics we fitted the function

$$P(T\_initiation>t) = \exp(-(t-t0)/T)$$

(solid black line) to the experimental observations, where the two unknown parameters, t0—an arbitrary time offset of the start of the observations, and T—the time constant of the Poisson process, were estimated using a least squares minimisation procedure (Fig. 2F). The fit describes the observations with $R^2 = 98\%$, suggesting that RNF213 coat initiation is indeed a Poisson process. The estimated time constant T is $27 \pm 1$ min, confirming the proposition that coat initiation is a rare event with an expectation of occurring once every 27 min.

### Evolutionary analysis of simian RNF213 genes

To examine the evolutionary selective pressures on RNF213, we collected orthologous sequences from simian primate species via a blastn search (Altschul et al, 1997) of NCBI's non-redundant nucleotide database using the human RNF213 ORF sequence as query (15,624 bp; from Genbank NM_001256071). We filtered the resulting blast output so that for each simian primate species, we retained only a single match, choosing the one with the highest bit score. Matching sequences were blasted against a local database of all human RefSeq sequences, confirming that RNF213 was indeed the best match rather than a paralogous gene. We extracted ORF sequences from each database entry and generated an in-frame alignment (Dataset EV1) of all 24 ORFs (including the human query sequence) using the PRANK algorithm (Löytynoja, 2020).

We used this alignment to generate a maximum likelihood phylogenetic tree with PHYML (Guindon et al, 2010) using the GTR nucleotide substitution model, estimating the proportion of invariable sites, shape of the gamma distribution and nucleotide frequencies. We then used the alignment and tree as input for PAML's codeml algorithm (Yang, 2007), which uses a maximum likelihood approach to estimate whether evolutionary models that allow a subset of codons to evolve under positive selection (e.g., "model 8") fit the data better than similar null models that allow only neutral and purifying selection (e.g., "model 8a", "model 7"). We compared two pairs of evolutionary models: model 8 versus 8a, and model 8 versus 7, in each case determining twice the difference in log-likelihoods and comparing that with a chi-squared distribution to generate a p-value (Yang, 2007). The results we present here are from codeml analyses that use the following parameters: initial_omega=0.4, codon_model=2, cleandata=0 (Appendix Table S1). However, we have also confirmed that results are robust to the use of alternative parameter choices with all four combinations of initial_omega=0.4 or 3, codon_model=2 or 3. Codeml also runs a Bayes Empirical Bayes analysis to determine the posterior probability that each codon evolves under positive selection. We used an arbitrary BEB probability threshold of ≥90% as a cutoff to report the residues most likely to be rapidly evolving.

We also checked the alignment for signs of recombination, which can lead to false evidence for positive selection (Anisimova et al, 2003). We used the GARD algorithm (Pond et al, 2006), allowing site-to-site rate variation under the general discrete model with 3 rate classes. GARD did find recombination in our RNF213 alignment (evidence ratio >100), suggesting that 6 segments of the alignment each have a different phylogeny. However, closer inspection revealed that the tree topologies produced by GARD for each segment (Fig. EV3A) roughly follow the known species tree (Perelman et al, 2011) and are only subtly different, with the exceptions of segments 1 and 2, where olive baboon and silvery gibbon

are wrongly placed; these two species have missing data in these segments (large alignment gaps due to incomplete gene predictions). As a result, GARD cannot accurately place them on a tree based on segments 1 and 2, but this also means that they cannot contribute to false evidence of positive selection in these regions.

Due to GARD's finding of possible recombination, we sliced the alignment into the 6 segments defined by GARD, trimmed each to remove partial codons at the start/end. We then analysed each segment separately for signatures of positive selection, using codeml as described for the full alignment above. In the first two segment alignments, we removed sequences contained only gaps (i.e., olive baboon and silvery gibbon). Codeml found robust evidence for positive selection in 5 of the 6 segment alignments i.e., in all segments except the most C-terminal (Dataset EV1, Appendix Table S1, Fig. EV3B). Our results are robust to the use of alternative parameter combinations.

### Statistical analysis

All data were tested for statistical significance with Prism software (GraphPad Prism 9) or Python (Python 3.6). The paired two-tailed Student's *t*-test was used to test whether two samples originated from the same population. Differences between more than two samples were tested using a one-way analysis of variance (ANOVA). Unless otherwise stated, all experiments were performed at least three times, and the data were combined for presentation as a mean ± s.e.m. All differences not specifically indicated as significant were not significant ($P > 0.05$). Significant values are indicated as $*P < 0.05$, $**P < 0.01$, and $****P < 0.0001$. Statistical details, including sample size ($n$), are reported in the figures and figure legends.

## Data availability

The cryo-EM maps and the refined atomic model of human RNF213 have been deposited in the Electron Microscopy Data Bank and the Protein Data Bank under the accession codes. EMD 19653, EMD 19654, EMD 19655, EMD 19656, EMD 19657, EMD 19658, EMD 19659, and PDB 8S24.

The source data of this paper are collected in the following database record: biostudies:S-SCDT-10_1038-S44319-024-00280-w.

## Peer review information

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

## Acknowledgements

This work was supported by the Medical Research Council as part of United Kingdom Research and Innovation [U105170648], the Wellcome Trust [222503/Z/21/Z], the National Institutes of Health (U54 AI170792 (PI: Nevan Krogan)), and the Swiss National Science Foundation (SNSF) Postdoc Mobility grant (P400PB_191083). HSM is an Investigator of the Howard Hughes Medical Institute. We acknowledge Diamond for access and support of the cryo-EM facilities at eBIC, proposal BI31336, funded by the Wellcome Trust, MRC and BBSRC. We thank Johannes Schwab and Dari Kimanius for helpful advice on cryo-EM data processing.

## Author contributions

**Ana Crespillo-Casado**: Conceptualization; Investigation; Writing—original draft; Writing—review and editing. **Prathyush Pothukuchi**: Investigation. **Katerina Naydenova**: Investigation. **Matthew C J Yip**: Investigation. **Janet M Young**: Investigation. **Jerome Boulanger**: Investigation. **Vimisha Dharamdasani**: Investigation. **Ceara Harper**: Investigation. **Pierre-Mehdi Hammoudi**: Investigation. **Elsje G Otten**: Investigation. **Keith Boyle**: Investigation. **Mayuri Gogoi**: Investigation. **Harmit S Malik**: Supervision; Investigation; Writing—review and editing. **Felix Randow**: Conceptualization; Supervision; Funding acquisition; Writing—original draft; Writing—review and editing.

Source data underlying figure panels in this paper may have individual authorship assigned. Where available, figure panel/source data authorship is listed in the following database record: biostudies:S-SCDT-10_1038-S44319-024-00280-w.

## Disclosure and competing interests statement

The authors declare no competing interests.

# Expanded View Figures

**Figure EV1. RNF213 accumulates on phylogenetically distant pathogens.**

(A) Confocal micrographs of HeLa cells infected with *S.* Typhimurium for 4 h and *L. monocytogenes* or *T. gondii* infected for 6 h. Cells were stained with anti-RNF213 antibody. Scale bar 20 μm (magnification box scale bar; 5 μm). (B) Confocal micrographs representative of quantifications shown in Fig. 1E. MEFs stimulated with IFNγ as indicated, infected with Tomato-expressing *T. gondii* Type I RH or Type II Pru for 1 h and stained with anti-ubiquitin (FK2) antibody and DAPI. Regions marked with white borders in the main images are shown magnified on the right. Scale bar 80 μm (magnification box, scale bar; 20 μm). Source data are available online for this figure.

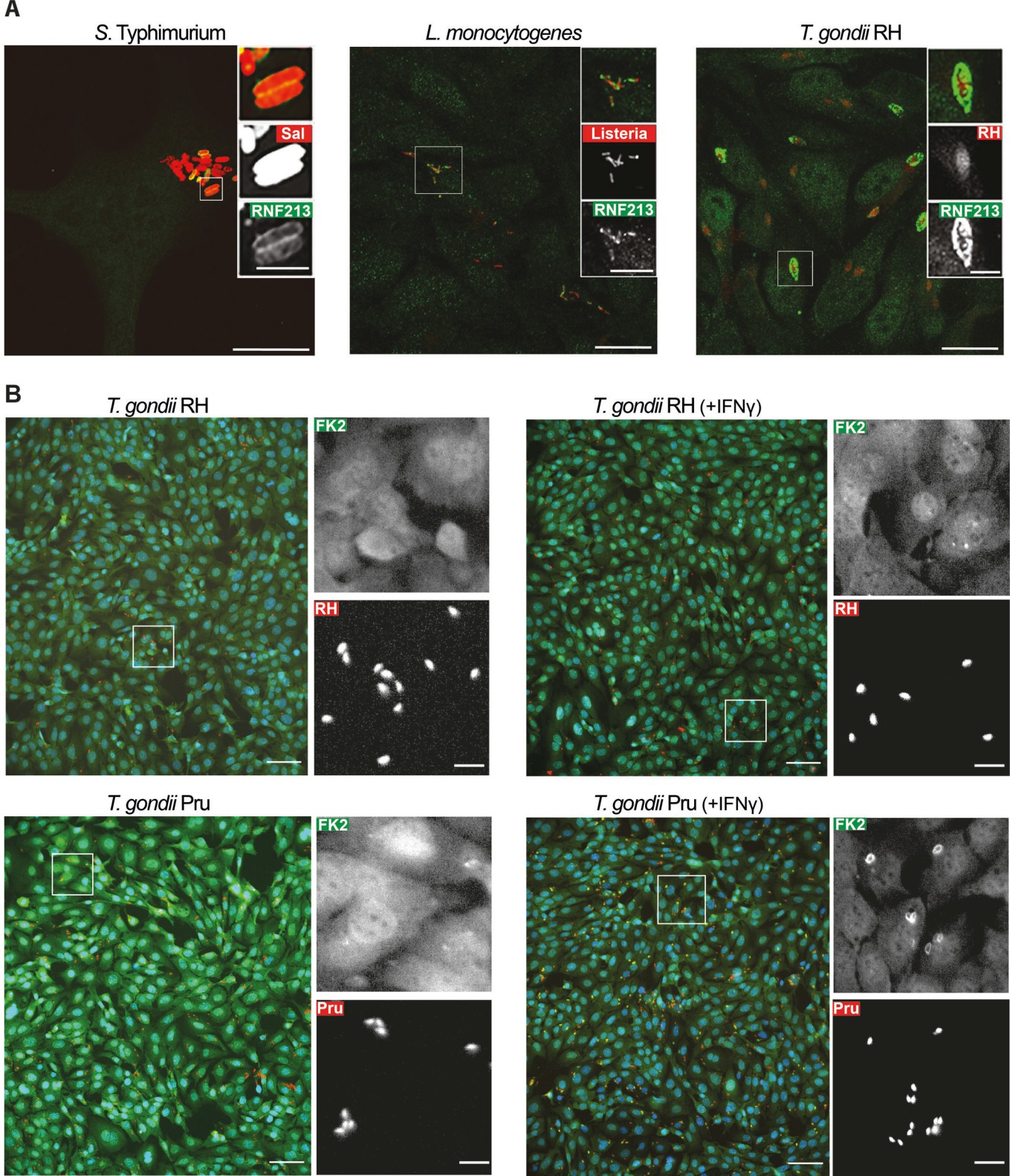

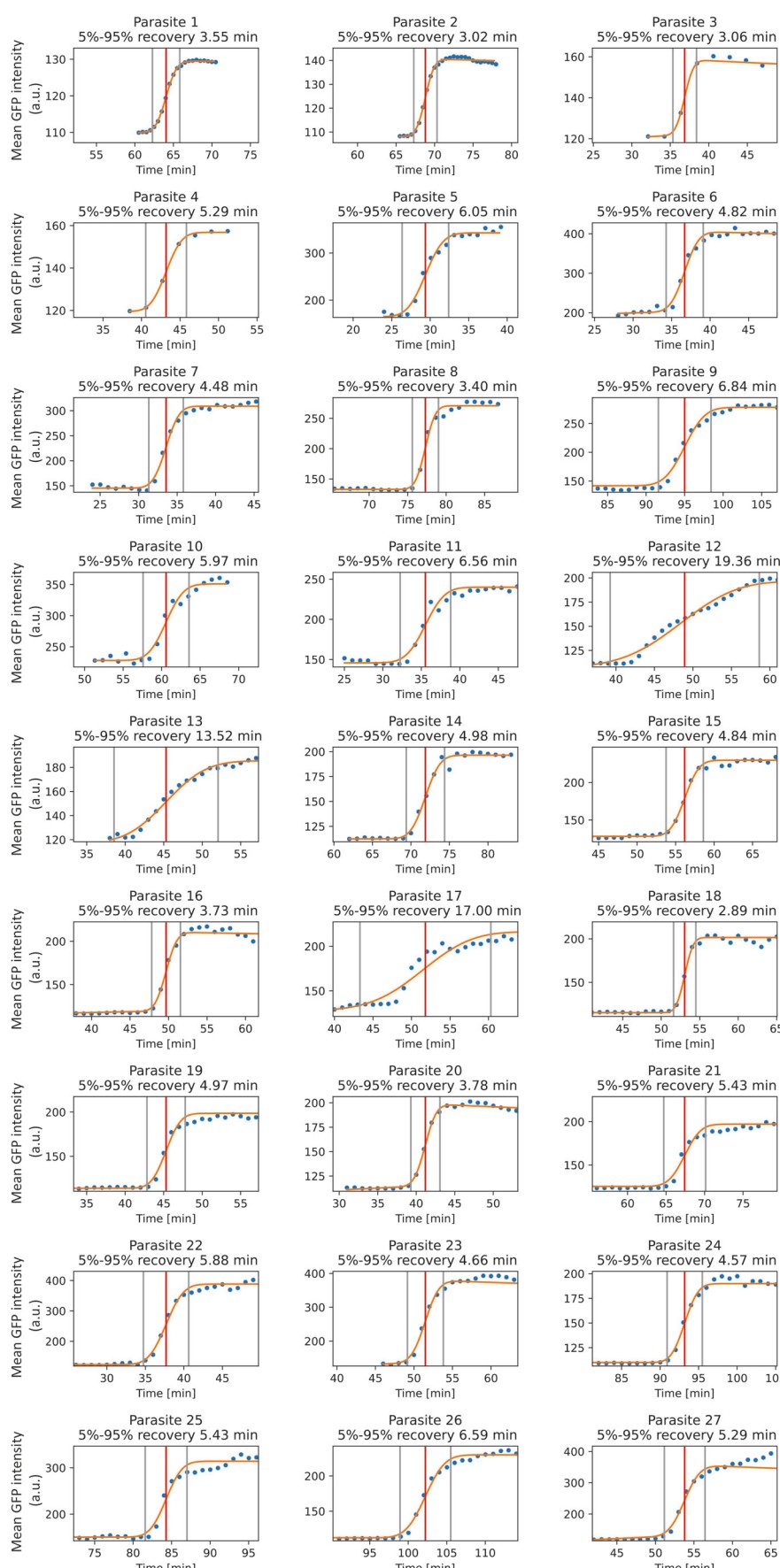

◀  **Figure EV2.   Kinetics of GFP-RNF213 accumulation on individual *T. gondii* vacuoles.**

Kinetics of GFP-RNF213 accumulation on *T. gondii* vacuoles. Dots represent the measured GFP intensity. An error function was fitted (orange line) to calculate the 5% and 95% of maximum coat formation (vertical grey lines) as well as the midpoint (vertical red line). Time required for coat completion (5% to 95% of maximum fluorescence) is annotated above the graphs. n = 6 biological experiments; parasites 1 and 2 from experiment 1; parasites 3 and 4 from experiment 2; parasites 5 to 10 from experiment 3; parasite 11 from experiment 4; parasites 12 to 21 from experiment 5; parasites 22 to 27 from experiment 6. Acquisition started at 60 min, 30 min, 23 min, 24 min, 30 min and 19 min p.i, respectively. Experiments 1 and 2 were acquired on a Nikon iSIM swept field high speed inverted microscope with a 100X super resolution Apo TIRF oil objective, experiments 3 to 6 on a Nikon X1 Spinning Disk inverted microscope with a 40x/1.3NA Oil lens. Source data are available online for this figure.

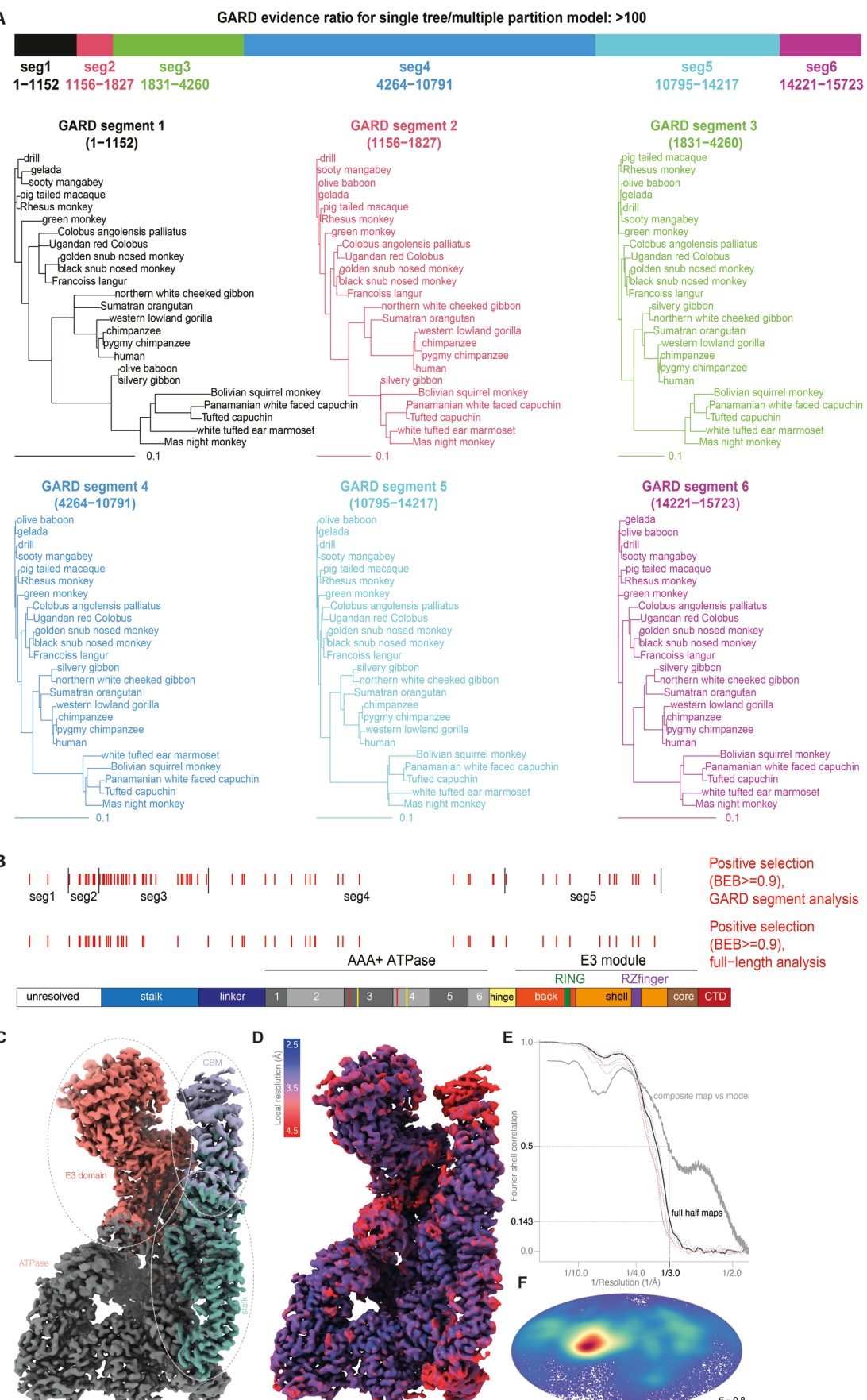

◄ **Figure EV3. Positive selection in GARD-identified RNF213 segments and cryo-EM analysis of RNF213.**

(A) The GARD algorithm detects evidence of possible recombination among RNF213 simian primate orthologs, with an evidence ratio of >100. Schematic of the 6 segments identified by GARD, and the phylogenies for each segment according to GARD. Differences between segment topologies are mostly subtle, in basal branches of the tree. (B) Repeat of codeml analysis, performed individually on each of the 6 GARD segment alignments. Segments 1–5 all showed evidence of positive selection (Appendix Table S1). Indicated with red tick marks are residues with high probability (Bayes Empirical Bayes posterior probability ≥0.9) of evolving under positive selection identified by GARD segment and full-length analysis, as indicated. Analysing the 6 GARD segments separately appears to increase the statistical power of codeml to identify rapidly evolving sites: across the 5 segments, a total of 86 sites have ≥90% posterior probability (Bayes Empirical Bayes) of evolving under positive selection, in contrast to 59 sites when analyzing the full-length alignment. (C) Composite map of RNF213 created by local refinement of individual color-coded domains. (D) Composite map of RNF213 coloured by local resolution, calculated using ResMap (Kucukelbir et al, 2014). (E) Fourier shell correlation between the two consensus half-maps (solid black line), and between the composite map and the refined atomic model (solid grey line), is plotted as a function of resolution. The Fourier shell correlations for each pair of locally refined half maps are shown with the dashed lines, coloured according to the domains in (A). (F) Orientation distribution of the RNF213 particles used in the consensus refinement shown on a Mollweide projection plot, coloured in blue to red from low to high density, and has an efficiency E = 0.8, calculated using cryoEF (Naydenova and Russo, 2017).

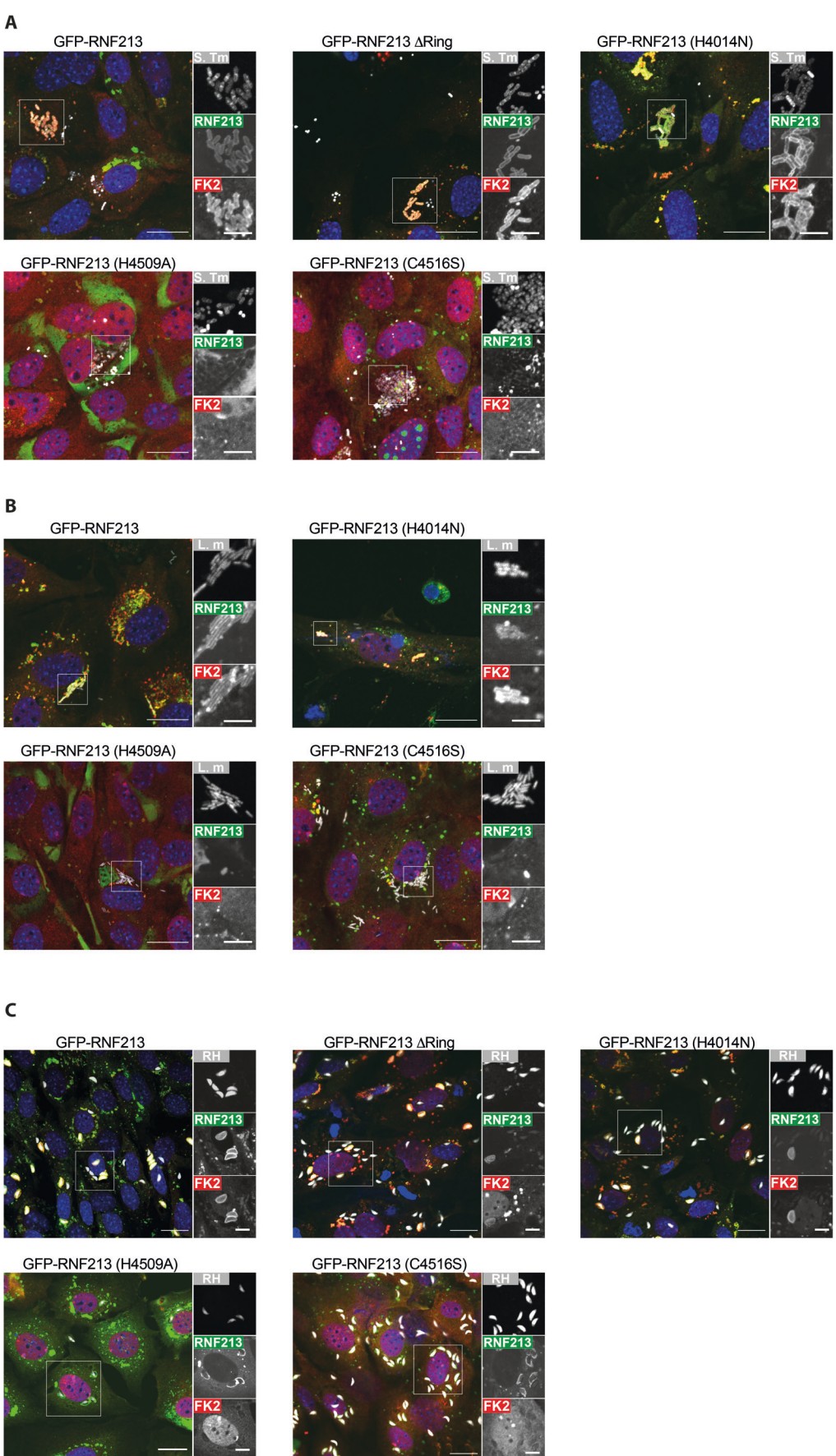

◄  **Figure EV4.  Enzymatic activity in RNF213 is required for coat formation on pathogens.**

Confocal micrographs representative of quantifications shown in Fig. 4D–F. RNF213$^{KO}$ MEFs complemented with the indicated GFP-RNF213 alleles and stained with anti-ubiquitin antibody (FK2) and DAPI at 4 h post-infection with mCherry-expressing *S. Typhimurium* (**A**), 6 h post-infection with mCherry-expressing *L. monocytogenes ΔActA* (**B**) and 1 h post-infection with Tomato-expressing *T. gondii* RH Type I strain (**C**). Regions marked with white borders in the main images are shown magnified on the right. Scale bar, 20 μm (magnification box scale bar, 10 μm). Source data are available online for this figure.

