## [Peer Review File · EMBO Reports]

Recognition of phylogenetically diverse pathogens through enzymatically amplified recruitment of RNF213

Ana Crespillo Casado, Prathyush Pothukuchi, Katerina Naydenova, Matthew Yip, Janet Young, Jerome Boulanger, Vimisha Dharamdasani, Ceara Harper, Pierre-Mehdi Hammoudi, Elsje Otten, Keith Boyle, Mayuri Gogoi, Harmit Malik, and Felix Randow

Corresponding author(s): Felix Randow (randow@mrc-lmb.cam.ac.uk)

Review Timeline:

Submission Date:	25th Jul 24
Editorial Decision:	26th Jul 24
Revision Received:	4th Sep 24
Editorial Decision:	12th Sep 24
Revision Received:	19th Sep 24
Accepted:	20th Sep 24

Editor: Achim Breiling

Transaction Report: Please note that the manuscript was transferred from another journal where it was originally reviewed. Since the original reviews are not subject to EMBO's transparent review process policy, they cannot be published.

Dear Dr. Randow,

Thank you for transferring your revised manuscript to EMBO reports. I now went through the manuscript, the referee reports from the assessment of the revised manuscript at the previous journal (outside EMBO press) and your point-by-point response. As you know, two referees are satisfied with the revisions and support publication of the study, whereas two referees remain critical, also indicating novelty concerns.

However, after looking through your point-by-point response, I have decided to invite a final revised manuscript that addresses the remaining referee points as indicated in your point-by-point response. Please also provide a detailed final point-by-point response to these points.

- 1) a .docx formatted version of the final manuscript text (including legends for main figures, EV figures and tables), but without the figures included. Figure legends should be compiled at the end of the manuscript text.
- 2) individual production quality figure files as .eps, .tif, .jpg (one file per figure), of main figures and EV figures. Please upload these as separate, individual files upon re-submission.

- 3) a .docx formatted letter INCLUDING the reviewers' reports and your final point-by-point response to their comments. As part of the EMBO Press transparent editorial process, the point-by-point response is part of the Review Process File (RPF), which will be published alongside your paper.

- 4) a complete author checklist, which you can download from our author guidelines

(<https://www.embopress.org/page/journal/14693178/authorguide>). Please insert page numbers in the checklist to indicate where the requested information can be found in the manuscript. The completed author checklist will also be part of the RPF.

- 5) that primary datasets produced in this study (e.g. RNA-seq, ChIP-seq, structural and array data) are deposited in an appropriate public database. If no primary datasets have been deposited, please also state this in a dedicated section (e.g. 'No primary datasets have been generated and deposited'), see below.

The accession numbers and database should be listed in a formal "Data Availability" section (placed after Materials & Methods) that follows the model below. This is now mandatory (like the COI statement). Please note that the Data Availability Section is restricted to new primary data that are part of this study. This section is mandatory. As indicated above, if no primary datasets have been deposited, please state this in this section

Data availability

8) Regarding data quantification and statistics, please make sure that the number "n" for how many independent experiments were performed, their nature (biological versus technical replicates), the bars and error bars (e.g. SEM, SD) and the test used to calculate p-values is indicated in the respective figure legends (also for EV figures and all those in an Appendix). Please also check that all the p-values are explained in the legend, and that these fit to those shown in the figure. Please provide statistical testing where applicable. Please avoid the phrase 'independent experiment', but clearly state if these were biological or technical replicates. Please also indicate (e.g. with n.s.) if testing was performed, but the differences are not significant. In case n=2, please show the data as separate datapoints without error bars and statistics. See also: <http://www.embopress.org/page/journal/14693178/authorguide#statisticalanalysis>

If n<5, please show single datapoints for diagrams. Please add to each legend (main, EV and Appendix figures, where applicable) a 'Data Information' section explaining the statistics used or providing information regarding replicates and scales. See:

9) Please add scale bars of similar style and thickness to microscopic images, using clearly visible black or white bars (depending on the background). Please place these in the lower right corner of the images themselves. Please do not write on or near the bars in the image but define the size in the respective figure legend.

10) Please also note our reference format:

12) We now use CRediT to specify the contributions of each author in the journal submission system. CRediT replaces the author contribution section. Please use the free text box to provide more detailed descriptions and do NOT provide your final manuscript text file with an author contributions section. See also our guide to authors: <https://www.embopress.org/page/journal/14693178/authorguide#authorshipguidelines>

13) All Materials and Methods need to be described in the main text using our 'Structured Methods' format, which is required for all research articles. According to this format, the Materials and Methods section should include a Reagents and Tools Table (listing key reagents, experimental models, software, and relevant equipment and including their sources and relevant identifiers), uploaded as separate file, followed by a Methods and Protocols section in which we encourage the authors to

describe their methods using a step-by-step protocol format with bullet points, to facilitate the adoption of the methodologies across labs. More information on how to adhere to this format as well as downloadable templates (.doc) for the Reagents and Tools Table can be found in our author guidelines (section 'Structured Methods'):

14) Please add up to five keywords and order the manuscript sections like this, using these names:

Title page - Abstract - Keywords - Introduction - Results - Discussion - Methods - Data availability section - Acknowledgements - Disclosure and Competing Interests Statement - References - Figure legends - Expanded View Figure legends

15) Please make sure that all the funding information is also entered into the online submission system and that it is complete and similar to the one in the acknowledgement section of the manuscript text file.

Yours sincerely,

Point-by-point reply

Reviewer #1 (Remarks to the Author):

The authors answered my questions thoroughly and I appreciate the work that they put into making a new Fig. 2.

We are pleased that Reviewer 1 is fully satisfied by how we addressed their questions.

Reviewer #2 (Remarks to the Author):

Despite the addition of some new structural data, the revised manuscript and the authors' responses have not addressed my primary concerns regarding data integrity, approach, and novelty claims.

The reviewer feels that we have not addressed his/her "... concerns regarding data integrity, approach, and novelty claims."

Our previous point-to-point reply tackled these concerns extensively. For example, regarding novelty we wrote:

"... Our manuscript provides the following novel insights:

- enzymatically driven cooperative recruitment of RNF213 enables sensing of evolutionarily distant pathogens
- a detailed kinetic analysis of RNF213 coat formation, providing time constants for coat initiation ($T=27\pm 1\text{min}$) and coat expansion ($5.0\pm 2.4\text{min}$) (new data in Fig.2C-F and supplementary). The difference in the time constant of coat initiation and the time required for coat expansion explains why coats on *T. gondii* vacuoles are initiated from a single focus; rarity of coat initiation and rapid coat completion limit the likelihood of more than one initiation event occurring per vacuole
- identification of a cluster of fast evolving residues near the N-terminus of RNF213, indicative of PRR function for RNF213
- a novel cryo-EM structure of human RNF213, revealing a carbohydrate-binding domain in the previously unresolved N-terminus that encompasses the aforementioned cluster of positively selected residues and strongly supports the proposed PRR function of RNF213 (new data in Fig.3)
- a cryo-EM structure and AlphaFold model of the RZ finger (new data in Fig.4C), supported by mutational analysis, offering mechanistic insight into the catalytic mechanism of the novel RZ-family class of E3 ligases."

Previous work had demonstrated that human RNF213 is essential for IFN γ -activated cell-autonomous immunity to *Toxoplasma* in human cells. The previous version of this manuscript claimed that mouse Rnf213 was similarly important for host defense to *Toxoplasma* in mouse cells. These claims were based on the use of an Rnf213 KO cell line made in an immortalized mouse embryonic fibroblasts (MEFs) cell line. Due to the nature of Ko generation in immortalized MEFs, it is likely that the Rnf213 KO MEF cell line is clonal. It is well documented that clonal effects can result in artificial phenotypes. I pointed out in my original critique that the data presented by Crespillo Casado et al was surprising for the reasons briefly recaptured here: the original version of this paper claimed that Rnf213 KO MEFs were defective in controlling the replication of both type I and type II *Toxoplasma* strains. Importantly, type I *Toxo* is resistant to IFN γ -mediated ubiquitination in mouse cells, and therefore the data presented by Crespillo Casado et al would have meant that Rnf213 would be exerting its anti-*Toxoplasma* activity independent of ubiquitination. In their rebuttal the authors now acknowledge that they cannot reproduce these findings in primary (non-clonal) Rnf213 KO MEFs and therefore it must be concluded that mouse Rnf213 – at the very least – is not essential for host defense to *Toxoplasma* – and mouse Rnf213 may very well not be sufficient either.

While I applaud the authors for removing their non-reproducible *Toxoplasma* restriction data from the revised manuscript, the revised manuscript continues to use these immortalized Rnf213 MEF KO cells, although the authors have now established that these cells are prone to generate artificial data – at least in *Toxoplasma* studies. In Fig.1F-G the paper continues to show data that would suggest that mouse Rnf213 promotes ubiquitination of the *Toxoplasma* parasitophorous vacuole (PV) in mouse cells. Again, these data – if reproducible – would be surprising because they would suggest that IFN γ -induced PV ubiquitination does not lead to PV destruction. These data were again generated in the potentially clonal immortalized Rnf213 KO MEF cell line. As I stressed in my previous critique, it is essential to complement this single Rnf213 KO clonal cell line through ectopic expression of mouse Rnf213 to make sure that the phenotype is not an artifact. In their response the authors highlight the difficulties of working with the large sized ORF of Rnf213. While I appreciate the technical difficulties of these experiments, these experiments are doable and the authors themselves point out that they have done these types of experiments in the past when they complemented Rnf213 KO cells with ectopic RNF213 in Otten et al. (2021). Alternatively, the authors could use independent primary KO MEFs (generated from different embryos as biological variables) and proper co-isogenic WT controls. These types of experiments are essential to support the paper's claim that Rnf213 promotes *Toxoplasma* PV ubiquitination in mouse cells.

The reviewer is concerned about potential species-specific differences in the ability of human and murine RNF213 to restrict *Toxoplasma*, specifically whether murine RNF213 promotes *Toxoplasma* PV ubiquitination in mouse cells. While this is a valid question, our manuscript aims to address how RNF213 senses phylogenetically distant pathogens, not whether murine RNF213 is deployed against *Toxoplasma* as was previously reported for human RNF213.

During the previous round of revision, we therefore removed data on the genetic interaction of *Toxoplasma* with murine RNF213 from the manuscript. Due to an oversight Fig.1F/G were left in the manuscript, which (understandably) triggered further questions from Reviewer 2. We would like to apologize for the mistake and have now removed the panels as originally intended. Importantly, our conclusion that RNF213 senses phylogenetically distant pathogens through enzymatically amplified RNF213 recruitment remains entirely unaffected.

The authors state in the rebuttal that because mouse and human RNF213 are quite homologous to each other, expression of human RNF213 in mouse cells is a sound approach. Yet, at the same time the authors claim (without any experimental evidence) that protein sequence variation in human RNF213 relative to other primate and monkey RNF213 orthologous is functionally important. I don't think you can have it both ways.

Because human RNF213 complements LPS ubiquitination in mouse Rnf213 KO cells (Otten et al.), the authors argue in their rebuttal that therefore "the evolutionary distance between humans and mice has not affected core RNF213 function." Of course, this logic is flawed. Just because LPS recognition is conserved between mouse and human RNF213 it does not mean that every other function of this very large multi-functional protein is conserved, especially considering that some residues

in RNF213 are under strong positive selection which implies changes in function. As argued in my initial critique, there is no compelling reason to work with human RNF213 in mouse cells yet many reasons why this flawed approach may result in artificial results.

Notwithstanding positive selection in 59 of the 5241 aligned residues (Fig.3A), RNF213 is a well conserved protein, as is evident from the negative selection scores for most residues (Appendix Table EV1) and the striking structural similarities between human and murine RNF213 (Ahel eLife 2020, this study Fig.3). We have shown experimentally that human RNF213 complements the murine RNF213^{-/-}MEFs used here regarding the ubiquitylation of Gram-negative bacteria (Otten Nature 2021) and Gram-positive bacteria (this study, Fig.1D, 4E).

Regarding the use of human RNF213 in murine cells, and as explained in our previous point-to-point reply, the enormous size of the RNF213 (with a cDNA larger than our mitochondrial genome!) poses significant practical challenges, which deterred us from generating murine copies of already existing human constructs. Expressing genes from phylogenetic distant species is a frequently used method to reveal functional conservation. In our case, expressing human RNF213 in murine cells is merely a practical approach to a technical difficulty.

To what degree the findings reported in this paper are novel is up to debate. However, I would argue that the novelty of the findings should be reflected in the title of the paper. The current title of the paper is 'Recognition of phylogenetically diverse pathogens through enzymatically amplified recruitment of RNF213.' The title therefore re-states what has already been reported by several previous publications: previous work had already reported that RNF213 can recognize phylogenetically diverse pathogens (Gram-negative and Gram-positive bacteria, protozoan pathogens, and viruses); and previous work has also already shown that recognition of these pathogens requires enzymatic activity of RNF213. The title should be changed and reflect (a) novel finding(s) of the study.

The reviewer incorrectly states that previous publication have shown that RNF213 "... recognizes phylogenetically distant pathogens", while instead it was only reported that RNF213 restricts a multitude of pathogens. The question whether RNF213 directly recognizes phylogenetic different pathogens or merely acts downstream of specific receptors has not been asked, yet is at the center of this manuscript.

The novelty of our manuscript therefore comes from addressing the question posed in the abstract "...How the E3 ubiquitin ligase RNF213 can respond to phylogenetically distant pathogens, including Gram-negative Salmonella, Gram-positive Listeria, and eukaryotic Toxoplasma, remains unknown."

Along the same lines, the claim made in the rebuttal that "enzymatically driven cooperative recruitment of RNF213 enables sensing of evolutionarily distant pathogens" is a novel finding is at the very least a vast exaggeration. First of all it has already been shown that RNF213 senses many phylogenetically diverse pathogens: the protozoan Toxoplasma (PMID: 38376248, PMID: 38147552, PMID: 36154443); Gram positive Listeria in the cytosol (PMID: 34599178, PMID:

34804992), gram-positive in a vacuole (PMID: 38147552), viruses (PMID: 33420513, PMID: 34599178, PMID: 36917666,), gram-negative in the cytosol (PMID: 34012115), Gram-negative in a vacuoles (PMID: 36084633).

The reviewer again claims that previous publication have shown RNF213 ‘...senses phylogenetically diverse pathogens.’ This statement is not correct because previous publications have merely shown restriction by RNF213 without addressing how pathogens are detected, i.e. whether RNF213 is a pattern recognition receptor of unprecedentedly broad reactivity or acts as an effector molecule downstream of hypothetical pattern recognition receptors.

It has also already been shown that sensing requires RNF213 enzymatic activity in the case of *Toxoplasma* (PMID: 38147552) and gram-negatives (PMID: 34012115 – this group’s own work). Time-lapse microscopy data (which is the only data underlying the claim of ‘cooperative recruitment’) similar to what is shown in Fig.2 of this manuscript has also already been reported (see figure 6 of PMID: 38147552). And time lapse microscopy for RNF213 binding to cytosolic *Salmonella* was also already published by this group and similar conclusions were already made – to quote from Otten et al (PMID: 34012115) “Instant structured illumination microscopy (iSIM) revealed that RNF213 recruitment started focally and subsequently spread around the bacterium, indicating cooperative behaviour (Video1, Fig.4c).” Just calling the same thing ‘cooperarative recruitment’ instead of ‘ cooperative behaviour’ doesn’t make it novel. Yes, the current study provides a more detailed analysis of the video microscopy data but does not reveal fundamentally novel insights.

In Otten et al (Nature 2021) we reported on the cooperative recruitment of RNF213 to *Salmonella*. Subsequently it became clear that other phylogenetically distant pathogens are also restricted by RNF213 (many references in our manuscript), suggesting that if RNF213 was a PRR, it would be of unprecedentedly broad reactivity. An urgent need therefore exists to investigate how phylogenetically distant pathogens might be recognized by one single receptor. Answering this question requires comparison of multiple pathogens side-by-side, which is what we have done in this study and which has never been done before. The reviewer’s suggestion that we merely renamed ‘cooperative behaviour’ into ‘cooperative recruitment’ is therefore incorrect. Instead we propose that recognition of all pathogens under study by RNF213 occurs in two steps, a rate-limiting initiation step that is not diffusion controlled and subsequent cooperative incorporation of further RNF213 molecules for multi-directional growth of the coat, for which we provide a detailed quantitative analysis.

I am not saying that everything reported in the current manuscript is already known. However, as stated in my previous critique of the paper, the genuinely novel finding of the original version of paper – in my view - is the identification of signatures of positive selection in human RNF213. While this is an exciting starting point for further exploration, the paper lacks any experimental evidence demonstrating that any of these signatures are functionally important. This type of functional analysis is typically expected of a high impact paper in this field. The revisions do not provide any new data demonstrating functional relevance and therefore the computational analysis on its own is only an incremental advance.

We disagree with the reviewer's opinion on novelty. Although not all questions have been answered, progress has been made. While the reviewer claims that the identification of positive selection in RNF213 is the only novel finding, we think that the following points are novel (and worth reporting):

- enzymatically driven cooperative recruitment of RNF213 enables sensing of evolutionarily distant pathogens
- a detailed kinetic analysis of RNF213 coat formation, providing time constants for coat initiation ($T=27\pm 1\text{min}$) and coat expansion ($5.0\pm 2.4\text{min}$) (new data in Fig.2C-F and supplementary). The difference in the time constant of coat initiation and the time required for coat expansion explains why coats on *T. gondii* vacuoles are initiated from a single focus; rarity of coat initiation and rapid coat completion limit the likelihood of more than one initiation event occurring per vacuole
- identification of a cluster of fast evolving residues near the N-terminus of RNF213, indicative of PRR function for RNF213
- a novel cryo-EM structure of human RNF213, revealing a carbohydrate-binding domain in the previously unresolved N-terminus that encompasses the aforementioned cluster of positively selected residues and strongly supports the proposed PRR function of RNF213 (new data in Fig.3)
- a cryo-EM structure and AlphaFold model of the RZ finger (new data in Fig.4C), supported by mutational analysis, offering mechanistic insight into the catalytic mechanism of the novel RZ-family class of E3 ligases.

Similarly, new data incorporated into the revised manuscript reveals the presence of a domain with structural similarity to the CMB20 carbohydrate-binding module of a fungal glucosamylase. This is an intriguing observation. However, the paper does not validate that the CMB20-like domain in RNF213 has any carbohydrate-binding activity or identifies any ligand – microbial or otherwise. The conclusions drawn by the authors are not substantiated by any experimental follow-up studies

With the possible exception of LPS, no ligand for RNF213 has been identified so far in any study and for any pathogen. Identifying the CBM20 domain in RNF213 as a hotspot of positive selection (and therefore likely receptor domain) should hence be considered significant progress.

please provide the sequence of the Rnf213 KO allele in MEFs. What was the sgRNA used to make the KO? Is it possible splice variants are still being expressed that may have functional effects?

The MEFs used here carry frameshifts in exon 31. If exon skipping occurs, multiple exons (i.e. exons 31-35) would need to be skipped to maintain an open reading frame. We therefore consider the existence of functional splice variants unlikely.

An additional comments regarding the authors' response:

- The authors argue that they cannot monitor restriction of *Toxoplasma* by DOX-inducible RNF213 because the experiments last for 10 – 15 days and DOX over this timeframe causes toxicity. However, RNF213 mediated host defense can be assessed by quantifying parasites per vacuole by IF at 24 hours post infection (see PMID: 36154443). Therefore, using their DOX-inducible constructs to assess anti-parasitic effects is doable.

This comment refers back to old data no longer present in the manuscript.

Reviewer #3 (Remarks to the Author):

I very much appreciate the authors' efforts to address my comments experimentally. The novel data, in particular the quantitative data on RNF213 coat formation on *T.gondii* vacuoles and the cryoEM structure of human RNF213 with CBM20 domain, significantly strengthen the manuscript which can now be accepted for publication. As a final remark: you might want to mention the new cryoEM structure explicitly in the abstract. Congratulations on the outstanding work - Francis Impens.

We are delighted that Reviewer 3 supports publication of our manuscript, which he/she considers significantly strengthened through the inclusion of the newly solved cryo-EM structure of human RNF213 and the addition of quantitative data on RNF213 coat formation.

Reviewer #4 (Remarks to the Author):

The revised manuscript presents two major updates: 1) quantitative analysis of RNF213 targeting and amplifying on the *Toxoplasma* vacuole-containing member (new Figure 2 C-F); 2) cryo-EM data on human RNF213 revealing a CBM domain in the N-terminus.

We are happy that the newly added quantitative analysis of RNF213 coat formation and the newly added high-resolution cryo-EM structure of RNF213 leading to the discovery of the previously unknown CBM20 domain in RNF213 are recognized by Reviewer 4 as 'major updates'.

While the current version of the manuscript resolves some of my previous concerns, there are still major issues that must be addressed, rendering it unsuitable for publication in its present form. Detailed comments are listed below:

1. Although the authors have determined the structure of human RNF213 and identified the CBM domain in the previously unresolved N-terminus region, their argument that RNF213 acts as a bona fide PRR serving as a bona fide PRR because of the positive selection in the CBM domain is too speculated and requires experimental evidence.

We agree with the reviewer that strong circumstantial evidence from evolutionary analysis does not provide direct proof of RNF213 serving PRR function. We therefore avoided any overstatements in our manuscript by merely concluding that our data are '...consistent with' RNF213 serving as a PRR.

Is there any prior study that suggests CBM is involved in pathogen recognition?

We thank the reviewer for asking about the CBM20 domain. We now realize that we may not have sufficiently explained why strong positive selection in a carbohydrate binding domain adds to the likelihood of RNF213 serving PRR function: a) pathogen detection through carbohydrate binding domains is a well-established concept and of particular appeal in the otherwise carbohydrate-devoid cytosol and b) the notorious diversity of microbial carbohydrates fits well with the pronounced positive selection observed in the CBM20 domain of RNF213.

In addition, since the truncation of the entire N-terminus (N586) leads to aggregation, it suggests CBM domain may serve as a signal anchor for correct trafficking of RNF213 or maintaining the oligomerization status of RNF213 rather than substrate binding. To support the current conclusion, experiments with point mutations in the CBM domain need to be carried out.

To address the reviewers request, we mutated residues in the CBM20 domain likely involved in carbohydrate recognition (F391, F401, W413, H421, Y422, Y462). However, we observed that mutant proteins were still able to detect pathogens (new data, Appendix Fig.S1), an ambiguous result that, unfortunately, neither proves nor refutes a role of the CBM20 domain in pathogen recognition. Further studies on the role of the CBM20 domain will be needed to establish its precise function.

2. Based on the authors' responses to all reviewers, it appears that the focus of this manuscript is on RNF213-mediated pathogen recognition, which has led to the omission of the Toxoplasma plaque assay data. Nonetheless, if the authors assert that RNF213 is a bona fide PRR, it remains unclear how exactly RNF213 recognizes the evolutionally distinct pathogens. Moreover, new Figures 2C to 2F indicate that the initiation of RNF213 recruitment and the amplification of the coating signal are two separate molecular events. Since the experiments examining RNF213 mutations were all conducted at a single time point, it is difficult to determine if these mutants have defects in pathogen recognition or in the amplification of RNF213 coating signal.

We thank the reviewer for the above succinct summary. Initiation of RNF213 recruitment and the enzymatic amplification of the RNF213 coat are indeed two separate events – the former relies on rate-limiting initiation events, while the latter requires the cooperative incorporation of further RNF213 molecules. However, as can be concluded from the kinetic analysis of RNF213 recruitment to Toxoplasma (Fig.2), defects in either pathogen recognition or subsequent incorporation of RNF213 monomers into the coat will result in the absence of RNF213 coats. We therefore think that the reviewer's suggestion of analyzing multiple time points with established methodology is unlikely to provide further mechanistic insights

Minor comments:

1. I have come to realize that the cells used in this study express RNF213 under doxycycline-induced conditions. Given this, performing all experiments on IFN γ -stimulated cells is misleading. Since IFN γ is known to induce the expression of RNF213 in human cells, as shown in PMID 36154443 and PMID38147552, the effect of IFN γ induction is not evident in this study (although the current work mainly carried out in murine fibroblasts). So, I suggest removing all data related to IFN γ -stimulated cells from this manuscript.

IFN γ is known to affect cells in many aspects, not only by controlling RNF213 levels. Regarding the results reported here, there seems to be a misunderstanding – most experiments were not performed in the presence of IFN γ (for example, those in Fig.2, 3, 4, or 5), while experiments involving IFN γ (Fig.1B, E) also contain conditions without cytokine treatment to enable direct comparison.

2. Figure 2D, please clarify whether a synchronized invasion assay was performed (see PMID 20435700 as an example). If not, the observed difference in RNF213 coat formation among individual parasites could be attributed to variations in the timing of parasite invasion. The time frame for Toxoplasma to invade host cells can range from 10 minutes to 2 hours after the parasites are introduced to the cells.

Synchronized spin infections were performed. To further clarify the method, the section "Life-cell microscopy" in Materials & Methods has been updated. We thank the reviewer for pointing out the ambiguity in the previous version of our manuscript.

2. Figure 3A and 3B appear identical, with the only difference being the CBM labeled in Figure 3B. It would be nice to combine them into one figure.

3.

We deliberately created two cartoons as the panels have distinct purposes. Fig3A displays sites of positive selection, mapped onto the domain structure of RNF213 as it was known before our study, thus highlighting the need for further structural analysis. Fig3B is supposed to guide the reader in understanding Fig.3C-G.

Dear Dr. Randow,

Thank you for the submission of your further revised manuscript to our editorial offices. I now went through your final p-b-p-response and the revised manuscript and consider the remaining points of referees #2 and #4 as adequately addressed.

Before we can proceed with formal acceptance, I have these final editorial requests:

- Please add the word 'Abstract' above the abstract.

- Please make sure that the number "n" for how many independent experiments were performed, their nature (biological versus technical replicates), the bars and error bars (e.g. SEM, SD) and the test used to calculate p-values is indicated in the respective figure legends. Please also check that all the p-values are explained in the legend, and that these fit to those shown in the figure. Please provide statistical testing where applicable. Please avoid the phrase 'independent experiment', but clearly state if these were biological or technical replicates. Please also indicate (e.g. with n.s.) if testing was performed, but the differences are not significant. In case $n=2$, please show the data as separate datapoints without error bars and statistics. See also: <http://www.embopress.org/page/journal/14693178/authorguide#statisticalanalysis>

If $n < 5$, please show single datapoints for diagrams. Could statistics also be provided for the diagrams in panels 1B and 1E? Moreover:

- Please show the stats for all diagrams as in panel 1B (with a line indicating which datasets have been tested).

- Please provide exact p values in the legends of figures 1d; 3h-j; 4a-b, d-f.

- Please note that in figure 3i; there is a mismatch between the annotated p values in the figure legend and the annotated p values in the figure file that should be corrected.

- Please note that information related to n is missing in the legend of figure 2e.

- Please define the error bars in the legends of figures 2e.

- Please add scale bars of similar style and thickness to all microscopic images (main, EV and Appendix figures), using clearly visible black or white bars (depending on the background). Please place these in the lower right corner of the images themselves. Please do not write on or near the bars in the image but define the size in the respective figure legend. Presently, several scale bars are too thin or have text nearby. Moreover, the scale bar in figure EV 1a needs to be defined in the legend. Please also add scale bars to the magnification boxes in Figs. 1, EV1 and EV4.

- Please add a title for each EV figure to the EV figure legends.

- Per journal policy, we do not allow 'results not shown', which is stated in the manuscript (page 28). All data referred to in the paper should be displayed in the main or Expanded View figures, or an Appendix. Thus, please add these data (or change the text accordingly if these data are not central to the study). See:

<https://www.embopress.org/page/journal/14693178/authorguide#unpublisheddata>

- Please provide the Appendix as pdf file.

- Please make sure that the Appendix Tables are called out as 'Appendix Table Sx'. There are callouts for "Table Sx" in the manuscript text (page 29). Please check.

- Please provide the legends for Dataset EV1 as a readme text file ZIPed together with the Fasta file.

- The two movie files (videos) need to be named 'Movie EV1' and 'Movie EV2'. Please also use these names for their callouts. Moreover, please upload these files zipped up together with a readme text file containing the respective legend.

- Then, please remove the 'Further legends' section from the manuscript text file.

- Table EV1 is a dataset. Please upload this as dataset file (named 'Dataset EV2') with its legend on the first TAB of the excel file. Please also update the call-out for this table accordingly and remove the legend for 'Table EV1' from the manuscript text file.

- Please add a paragraph titled 'Biosafety' to the methods section gathering all information on where and how biosafety-relevant experiments with pathogens were performed and that these were approved, and by whom (institution, government).

- Please remove the instructions and examples from the reagents and tools table.

- Please provide a final Data Availability Section (that should only mention externally deposited datasets) with specific URLs

(direct links) for the EMD 19653 - 19659 and PDB 8S24 datasets. Please make sure these are public and accessible latest upon publication of the manuscript.

- Thank you for providing the requested source data. Please upload the source data for the main figures as one folder per figure (with all files for one figure in one folder and ZIPed).

In addition, I would need from you uploaded separately:

Best,

The authors have addressed all minor editorial requests.

Dr. Felix Randow
MRC Laboratory of Molecular Biology
Division of Protein and Nucleic Acid Chemistry
Francis Crick Avenue
Cambridge CB2 0QH
United Kingdom

Dear Dr. Randow,

I am very pleased to accept your manuscript for publication in the next available issue of EMBO reports. Thank you for your contribution to our journal.

Yours sincerely,
